# ID$^3$: Identity-Preserving-yet-Diversified Diffusion Models for Synthetic Face Recognition

Jianqing Xu[*1],     Shen Li[*2],
Jiaying Wu[2], Miao Xiong[2], Ailin Deng[2], Jiazhen Ji[1],
Yuge Huang[#1], Guodong Mu[1], Wenjie Feng[2], Shouhong Ding[1], and Bryan Hooi[2]

[1]Tencent Youtu Lab
[2]National University of Singapore

## Abstract

Synthetic face recognition (SFR) aims to generate synthetic face datasets that mimic the distribution of real face data, which allows for training face recognition models in a privacy-preserving manner. Despite the remarkable potential of diffusion models in image generation, current diffusion-based SFR models struggle with generalization to real-world faces. To address this limitation, we outline three key objectives for SFR: (1) promoting diversity across identities (interclass diversity), (2) ensuring diversity within each identity by injecting various facial attributes (intra-class diversity), and (3) maintaining identity consistency within each identity group (intra-class identity preservation). Inspired by these goals, we introduce a diffusion-fueled SFR model termed ID$^3$. ID$^3$ employs an ID-preserving loss to generate diverse yet identity-consistent facial appearances. Theoretically, we show that minimizing this loss is equivalent to maximizing the lower bound of an adjusted conditional log-likelihood over ID-preserving data. This equivalence motivates an ID-preserving sampling algorithm, which operates over an adjusted gradient vector field, enabling the generation of fake face recognition datasets that approximate the distribution of real-world faces. Extensive experiments across five challenging benchmarks validate the advantages of ID$^3$. Code is released at: `https://github.com/hitspring2015/ID3-SFR`.

## 1 Introduction

With the introduction of various regulations restricting the use of large-scale facial data in recent years, such as GDPR, synthetic-based face recognition (SFR) (Boutros et al., 2023) has received widespread attention from the academic community (Qiu et al., 2021; Wood et al., 2021; Wang et al., 2023). The goal of SFR is to generate synthetic face datasets that mimic the distribution of real face images, and use it to train a face recognition (FR) model such that the model can recognize real face images as effectively as possible.

There exist numerous efforts to address SFR, which can be categorized into *GAN*-based models and *diffusion* models. GAN-based models utilize adversarial training to learn to generate synthetic data for FR training. Recently, with the empirical advantages of diffusion models over GANs, many works have attempted to use diffusion models to generate synthetic face data in place of authentic data. However, the reported results by these state-of-the-art (SoTA) SFR generative models (Bae et al., 2023; Boutros et al., 2022; Kolf et al., 2023; Qiu et al., 2021; Boutros et al., 2023) show

---

[*]Equal first authors. The order was determined by `numpy.random.rand()`.
[#]Corresponding author.

significant degradation in the verification accuracy in comparison to FR models trained by authentic data. We deduce the degradation might be due to two reasons. First, while previous works adopt diffusion models, they operate in the original score vector field without injecting the direction with regards to identity information, which makes them unable to guarantee identity-preserving sampling. Second, they fail to consider the structure of face manifold in terms of diversity during sampling.

We thus argue that the crux of SFR is to automatically generate a training dataset that has the following characteristics: (i) *inter-class diversity*: the training dataset covers sufficiently many distinct identities; (ii) *intra-class diversity*: each identity has diverse face samples with various facial attributes such as poses, ages, etc; (iii) *intra-class identity preservation*: samples within each class should be identity-consistent. Also note that, critically, the SFR dataset generation process should be fully automated without manual filtering or introducing auxiliary real face samples.

To this end, in this paper, we propose a novel **ID**entity-preserving-yet-**D**iversified **D**iffusion generative model termed $ID^3$ and a sampling algorithm for inference. Jointly leveraging identity and face attributes as conditioning signals, $ID^3$ can synthesize diversified face images that conform to desired attributes while preserving intra-class identity. Specifically, $ID^3$ generates a new sample based upon two conditioning signals: a target face embedding and a specific set of face attributes. The target face embedding enforces identity preservation while face attributes enrich intra-class diversity. To optimize $ID^3$, we propose a new loss function that involves an explicit term to preserve identity. Theoretically, we show that with the addition of this term, minimizing the proposed loss function is equivalent to maximizing the lower bound of the likelihood of an adjusted conditional data log-likelihood. Consequently, this theoretical analysis motivates a new ID-preserving sampling algorithm that generates desired synthetic face images. To generate an SFR dataset, we further propose a new dataset-generating algorithm. This algorithm ensures inter-class diversity by solving the Tammes problem (Tammes, 1930), which maximally separates identity embeddings on the face manifold. In the meantime, it encourages intra-class diversity by perturbing identity embeddings randomly within prescribed areas. It works in conjunction with identity embeddings and diverse attributes to ensure inter-/intra-class diversity while preserving identity. Extensive experiments show that $ID^3$ outperforms other existing methods in multiple challenging benchmarks.

To sum up, our major contributions are listed as follows:

- **Model with Theoretical Guarantees**: We propose $ID^3$, an identity-preserving-yet-diversified diffusion model for SFR. Theoretically, optimizing $ID^3$ is equivalent to shifting the original data likelihood to cover ID-preserving data.

- **Algorithm Design**: Motivated by this theoretical equivalence, we design a novel sampling algorithm for face image generation, together with a face dataset-generating algorithm, which effectively generates fake face datasets that approximate real-world faces.

- **Effectiveness**: Compared with SoTA SFR approaches, $ID^3$ improves SFR performance by $\sim 2.4\%$ on average across five challenging benchmarks.

## 2    Problem Formulation

The scope of this paper is synthetic-based face recognition (SFR), which focuses on generating high-quality training data (i.e., face images) for FR models. Generally, we aim to address SFR by generating face images that conform to diverse facial attributes while preserving identity within each class, in an automated manner. Technically, we break down this objective into the following two research questions (RQs) to be answered:

- **RQ1**: How can we effectively train a SFR generative model that preserves identity within each class, while boosting inter-class and intra-class diversity?

- **RQ2**: Once the generative model is trained, what sampling strategy can be employed to generate a synthetic face dataset that enables state-of-the-art face recognition models to perform well on real face benchmarks?

The rest of the paper aims to answer these two questions, respectively, in order to improve synthetic face recognition performance.

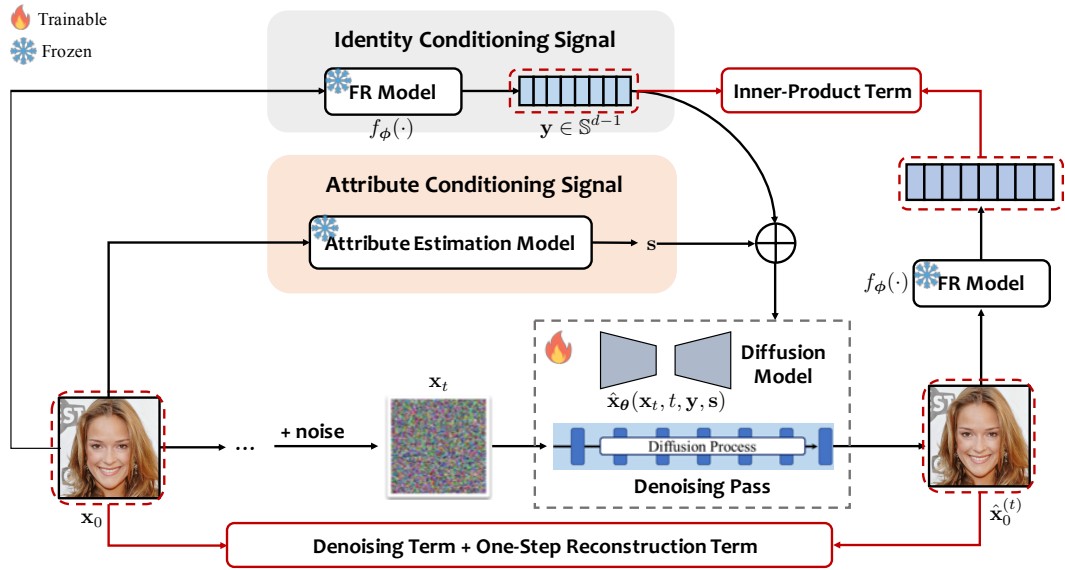

Figure 1: The forward pass of $\text{ID}^3$ in terms of loss computation. Given an image, its face attributes, and its face embedding, $\text{ID}^3$ obtains the image's noised version after $t$ diffusion steps and employs a denoising network to denoise it. This denoising process is conditioned on the predicted attributes and the ID embedding. Optimization proceeds by minimizing a loss function comprised of a denoising term, a one-step reconstruction term, an inner-product term, and a constant.

## 3  Methodology

We propose $\text{ID}^3$, a conditional diffusion model that generates diverse yet identity-preserving face images. $\text{ID}^3$ solves RQ1 by introducing two conditioning signals (identity embeddings and face attributes) into a diffusion model which is trained using a novel loss function. The loss function, together with identity embeddings, ensures intra-class identity preservation, while generation upon various face attributes give rise to intra-/inter-class diversity of face appearances. Our theoretical result regarding this loss function leads to an ID-preserving sampling algorithm and, further, an effective dataset-generating algorithm.

**Notations.** Throughout the rest of the the paper, we let $\mathcal{D}$ denote a real face dataset that contains face images $\mathbf{x}_0 \in \mathbb{R}^{H \times W \times 3}$. Let $\mathbf{y}$ denote a desired identity embedding and $\mathbf{s}$ be face attributes.

### 3.1  Diffusion Models

We build up our generative model, $\text{ID}^3$, upon denoising diffusion probabilistic models (diffusion models for short) (Ho et al., 2020; Song et al., 2022; Rombach et al., 2022) as they empirically exhibit SoTA performance in the field of image generation. Diffusion models can be seen as a hierarchical VAE whose optimization objective is to minimize the KL divergence between the true data distribution and the model distribution $p_{\boldsymbol{\theta}}$, which is equivalent to minimizing the expected negative log-likelihood (NLL), $\mathbb{E}_{\mathbf{x} \sim \mathcal{D}}[-\log p_{\boldsymbol{\theta}}(\mathbf{x})]$. However, directly minimizing the expected NLL is intractable, therefore diffusion models instead maximize its evidence lower bound (ELBO), where the ELBO term can further simply to a denoising task with several model assumptions:

$$\log p(\mathbf{x}) \geq \underbrace{\mathbb{E}_{q(\mathbf{x}_{1:T}|\mathbf{x}_0)} \left[ \log \frac{p(\mathbf{x}_{0:T})}{q(\mathbf{x}_{1:T}|\mathbf{x}_0)} \right]}_{\text{ELBO}}$$

$$= \mathbb{E}_{q(\mathbf{x}_1|\mathbf{x}_0)} \left[ -\frac{1}{2} \|\mathbf{x}_0 - \hat{\mathbf{x}}_{\boldsymbol{\theta}}(\mathbf{x}_1, 1)\|_2^2 \right] - \frac{1}{T-1} \sum_{t=2}^{T} \mu_t \|\mathbf{x}_0 - \hat{\mathbf{x}}_{\boldsymbol{\theta}}(\mathbf{x}_t, t)\|_2^2 \tag{1}$$

where $\mu_t := \frac{T-1}{2\sigma_q^2(t)} \cdot \frac{\bar{\alpha}_{t-1}(1-\alpha_t)^2}{(1-\bar{\alpha}_t)^2}$, $\bar{\alpha}_t = \prod_{\tau=1}^{t} \alpha_\tau$. Specifically, given a sample $\mathbf{x}_0$ (or interchangeably, $\mathbf{x}$) from the image distribution, a sequence $\mathbf{x}_1, \mathbf{x}_2, ..., \mathbf{x}_T$ of noisy images is produced by

progressively adding Gaussian noise according to a variance schedule $\alpha_1, ..., \alpha_T$. This process is called the forward diffusion process $q(\mathbf{x}_t|\mathbf{x}_{t-1})$. At the final time step, $\mathbf{x}_T$ is assumed to be pure Gaussian noise: $\mathbf{x}_T \sim \mathcal{N}(0, I)$. The objective is to train a denoising network $\hat{\mathbf{x}}_{\boldsymbol{\theta}}$ that is able to predict the original image from the noisy image $\mathbf{x}_t$ and the time step $t$. To sample a new image, we sample $\mathbf{x}_T \sim \mathcal{N}(0, I)$ and iteratively denoise it, producing a sequence $\mathbf{x}_T, \mathbf{x}_{T-1}, ..., \mathbf{x}_1, \mathbf{x}_0$. The final image, $\mathbf{x}_0$, should resemble the training data.

Although the naive diffusion models are powerful in generating images, they do not deliver the promise of generating face images of the same identity (i.e. identity preservation) without direct corresponding information; nor are they aware of diverse desired facial attributes during inference. To achieve *intra-class diversity* and *intra-class identity preservation*, we would like to gain control of generating desired identities, each of which exhibits various attributes, including poses, ages and background variations. Hence, our aim is to design a diffusion model that conditions on specific identities and attributes throughout the generation of face images.

## 3.2  ID$^3$ as Conditional Diffusion Models

We propose a conditional diffusion model, ID$^3$ (see Figure 1 for details). Specifically, we extend the denoising network by conditioning it on two sources of signals: identity signals $\mathbf{y}$ and face attribute signals $\mathbf{s}$. The identity signals capture discernible faces in generated images, whereas face attribute signals specify the identity-irrelevant attributes, including poses, ages, etc. We introduce how to obtain these two conditioning signals, respectively, in the next two subsections.

### 3.2.1  Identity Conditioning Signal

To obtain identity conditioning signals, we assume access to a pretrained face recognition model $f_{\boldsymbol{\phi}} : \mathbb{R}^{H \times W \times 3} \mapsto \mathbb{S}^{d-1}$, which maps the domain of face images to a feature space $\mathbb{S}^{d-1}$. This mapping $f_{\boldsymbol{\phi}}$ is parameterized by the learnable parameter $\phi$, which is obtained by training the model on a real face dataset in the face recognition task. We follow the latest advancement of face recognition by setting the output space to be a unit hypersphere $\mathbb{S}^{d-1}$. Then, given a face image $\mathbf{x}_0$ drawn from the dataset $\mathcal{D}$, we obtain its identity embedding $\mathbf{y} \in \mathbb{S}^{d-1}$ by feeding it into a face recognition model $f_{\boldsymbol{\phi}}$: $\mathbf{y} = f_{\boldsymbol{\phi}}(\mathbf{x}_0)$, which serves as the identity conditioning signals for ID$^3$.

### 3.2.2  Face Attribute Conditioning Signal

Face attributes capture identity-irrelevant information about face images, such as age, face poses, etc. To obtain face attribute as conditioning signals, we employ pretrained attribute predictors (Serengil and Ozpinar, 2021) which output these attributes when given a face image as input. The pretrained attribute predictors are a collection of ad-hoc domain experts in age estimation and pose estimation. After obtaining each of these attribute values, $\mathbf{s}_{\text{age}} \in [0, 100]$, $\mathbf{s}_{\text{pose}} \in [-90°, 90°]^3$, we concatenate them as the overall attribute $\mathbf{s} = [\mathbf{s}_{\text{age}}, \mathbf{s}_{\text{pose}}]$ which is then fed into the diffusion model as conditioning signals.

## 3.3  Optimization Objective

Now the denoising network in Eq. (1) becomes $\hat{\mathbf{x}}_{\boldsymbol{\theta}}(\mathbf{x}_t, t, \mathbf{y}, \mathbf{s})$ that takes as input the noised $\mathbf{x}_t$, the time step $t$, and the conditioning signals $\mathbf{y}$ and $\mathbf{s}$. To optimize ID$^3$, we construct a training objective upon the ELBO of $\log p(\mathbf{x}|\mathbf{y}, \mathbf{s})$, ensuring that ID$^3$ generates identity-preserving yet diversified faces:

$$\min_{\boldsymbol{\theta}} \mathbb{E}_{(\mathbf{x}_0, \mathbf{y}, \mathbf{s}) \sim \mathcal{D}'} \left[ \mathcal{L}_{\boldsymbol{\theta}, \boldsymbol{\phi}}(\mathbf{x}_0, \mathbf{y}, \mathbf{s}) \right] \tag{2}$$

Here, $\boldsymbol{\theta}$ is the learnable parameter of the denoising network and the datapoint-wise loss is given by

$$\mathcal{L}_{\boldsymbol{\theta}, \boldsymbol{\phi}}(\mathbf{x}_0, \mathbf{y}, \mathbf{s})$$

$$= \mathbb{E}_{t \sim \mathcal{U}[2,T]} \left[ \underbrace{\mu_t \left\| \mathbf{x}_0 - \hat{\mathbf{x}}_0^{(t)} \right\|_2^2}_{\text{denoising term}} - \underbrace{\lambda_t \kappa_{\mathbf{x}_0} \mathbf{y}^T f_{\boldsymbol{\phi}}\left( \hat{\mathbf{x}}_0^{(t)} \right)}_{\text{inner-product term}} \right] + \mathbb{E}_{q(\mathbf{x}_1|\mathbf{x}_0)} \left[ \underbrace{\frac{1}{2} \left\| \mathbf{x}_0 - \hat{\mathbf{x}}_0^{(1)} \right\|_2^2}_{\text{one-step reconstruction term}} \right] + C$$

$$\tag{3}$$

| **Algorithm 1:** Training Algorithm | **Algorithm 2:** ID-Preserving Sampling Alg. |
|---|---|
| **Input:** The training face images $\mathbf{x}_0 \sim \mathcal{D}$; The pretrained face recognition model $f_\phi(\cdot)$. | **Input:** Denoising network $\hat{\mathbf{x}}_\theta$; recognition model $f_\phi$; conditioning signals $\mathbf{y}$ and $\mathbf{s}$. |
| **Output:** The denoising network $\hat{\mathbf{x}}_\theta$. | **Output:** A generated face $\mathbf{x}_0$ |
| Initialize $\mathcal{D}' \leftarrow \emptyset$ | $\mathbf{x}_T \leftarrow$ sample from $\mathcal{N}(0, I)$; |
| **for** $\mathbf{x}_0 \sim \mathcal{D}$ **do** | **for** $t \leftarrow T$ **to** $1$ **do** |
| $\quad \mathbf{y} \leftarrow f_\phi(\mathbf{x}_0)$; | $\quad$ Compute the score function $\nabla \log \tilde{p}(\mathbf{x}_t \vert \mathbf{y}, \mathbf{s})$ |
| $\quad \mathbf{s} \leftarrow \text{AttributePredictor}(\mathbf{x}_0)$; | $\quad\quad$ as in Eq. (7); |
| $\quad \mathcal{D}' \leftarrow \mathcal{D}' \cup \{(\mathbf{x}_0, \mathbf{y}, \mathbf{s})\}$; | $\quad$ Draw a Gaussian sample $\epsilon \sim \mathcal{N}(0, I)$; |
| **end** | $\quad$ Perform the update: |
| Solve Eq. (2) using batched Backpropagation algorithm with $\mathcal{D}'$; | $\quad\quad \mathbf{x}_{t-1} \leftarrow \mathbf{x}_t + \gamma \nabla \log \tilde{p}(\mathbf{x}_t \vert \mathbf{y}, \mathbf{s}) + \sqrt{2\gamma}\epsilon$; |
| | **end** |
| **return** $\hat{\mathbf{x}}_\theta$ | **return** $\mathbf{x}_0$ |

where $\hat{\mathbf{x}}_0^{(t)}$ is the output of the denoising network that takes as input the conditioning signals $\mathbf{y}, \mathbf{s}$, the time $t$ and the $t$-step noisified image $\mathbf{x}_t$:

$$\hat{\mathbf{x}}_0^{(t)} := \hat{\mathbf{x}}_\theta(\mathbf{x}_t, t, \mathbf{y}, \mathbf{s}). \tag{4}$$

Symbolically, $\hat{\mathbf{x}}_0^{(t)}$ denotes the denoised image predicted by the denoising network when given the $t$-step noisified $\mathbf{x}_t$, the time $t$ and the associated conditioning signals $\mathbf{y}, \mathbf{s}$. The coefficients, $\kappa_{\mathbf{x}_0}$ and $\lambda_t$ are scalars depending on $\mathbf{x}_0$ and $t$, respectively, and $C$ is a constant that does not depend on the learnable parameters $\boldsymbol{\theta}$. The specific value of $C$ will be elaborated in Appendix A.

To summarize, our proposed loss function consists of four terms: the one-step reconstruction term, the denoising term, the inner-product term, and a constant. Intuitively, the denoising term, along with the one-step reconstruction term, aims to improve the generative quality by denoising the $t$-step noisified face images while the inner-product term encourages the face embedding of the denoisified images to get close to the groundtruth identity embedding. To understand this loss function systematically, we theoretically find that minimizing this proposed loss function is equivalent to the maximization of the lower bound of an adjusted conditional log-likelihood over identity-preserving face images, which further leads us to an ID-preserving sampling algorithm.

**Theorem 3.1.** *Minimizing $\mathcal{L}$ with regard to $\boldsymbol{\theta}$ is equivalent to minimizing the upper bound of an adjusted conditional data negative log-likelihood $-\log \tilde{p}(\mathbf{x} \vert \mathbf{y}, \mathbf{s})$, i.e.:*

$$\min_{\boldsymbol{\theta}} \mathcal{L}(\mathbf{x}_0, \mathbf{y}, \mathbf{s}) \geq -\log \tilde{p}(\mathbf{x} \vert \mathbf{y}, \mathbf{s}) \tag{5}$$

*where*

$$\tilde{p}(\mathbf{x} \vert \mathbf{y}, \mathbf{s}) \propto p(\mathbf{x} \vert \mathbf{y}, \mathbf{s}) \cdot p(\mathbf{y}, \mathbf{s} \vert \mathbf{x})^{\frac{\sum_{t=2}^{T} \lambda_t}{T-1}} \tag{6}$$

*Proof.* The proof can be found in Appendix A. $\qquad\square$

**Remark.** We have just shown that our proposed loss is the upper bound of an adjusted conditional negative data log-likelihood. This adjusted likelihood $\tilde{p}(\mathbf{x} \vert \mathbf{y}, \mathbf{s})$ can be factorized into the original likelihood $p(\mathbf{x} \vert \mathbf{y}, \mathbf{s})$ and a reversed likelihood $p(\mathbf{y}, \mathbf{s} \vert \mathbf{x})$ with some positive power. We term it as "adjusted" since the original likelihood is discounted by the reversed likelihood. Intuitively, the reversed likelihood shifts the original likelihood such that the adjusted likelihood covers ID-preserving data, which is attributed to the inner-product term we introduce into the loss function in Eq. (2).

### 3.4 ID-Preserving Sampling

Theorem 3.1 provides insights for designing a novel sampling algorithm in the spirit of Langevin dynamics applied on the adjusted conditional likelihood $\tilde{p}(\mathbf{x}_t \vert \mathbf{y}, \mathbf{s})$. We note that Langevin dynamics can generate new samples from a probability density $p$ by virtue of its score function (i.e., the gradient of the logarithm of the probability density w.r.t. the sample, $\nabla_{\mathbf{x}} \log p$). Motivated by this observation, we aim to find the score function of the adjusted likelihood for sample generation. Specifically, taking the logarithm and the gradient w.r.t. $\mathbf{x}$ on both sides of Eq. (6) yields

$$\nabla \log \tilde{p}(\mathbf{x} \vert \mathbf{y}, \mathbf{s}) = \nabla \log p(\mathbf{x} \vert \mathbf{y}, \mathbf{s}) + \frac{\sum_{t=2}^{T} \lambda_t}{T-1} \nabla \log p(\mathbf{y}, \mathbf{s} \vert \mathbf{x}) \tag{7}$$

**Algorithm 3:** Synthetic Dataset Generation

---

**Input:** Denoising network $\hat{\mathbf{x}}_{\boldsymbol{\theta}}$; recognition model $f_{\boldsymbol{\phi}}$; the
    number of identities $N$.
**Output:** A synthetic dataset $\mathcal{D}_{\text{syn}}$.
$\mathcal{D}_{\text{syn}} \leftarrow \emptyset$;
Generate $\mathbf{w}_1, ..., \mathbf{w}_N \in \mathbb{S}^{d-1}$ by solving the Tammes
  problem;
**for** $i \leftarrow 1$ **to** $N$ **do**
    Generate $s_{1i}, ..., s_{mi} \sim \mathcal{U}[lb, ub]$;
    Calculate $\mathbf{Y}_i$ by solving the optimization problem $\mathbf{P}_i$ in
      Eq. (9);
    $\mathbf{y}_{i1}^*, ..., \mathbf{y}_{im}^* \leftarrow \text{unpack}(\mathbf{Y}_i)$;
    $\mathbf{s}_{i1}, ..., \mathbf{s}_{im} \leftarrow$ generate different attributes;
    $\mathcal{D}_i \leftarrow \emptyset$;
    **for** $j \leftarrow 1$ **to** $m$ **do**
      $\mathbf{x}_0 \leftarrow \text{Alg. 2}(\hat{\mathbf{x}}_{\boldsymbol{\theta}}, f_{\boldsymbol{\phi}}, \text{norm}(\mathbf{y}_{ij}^*), \mathbf{s}_{ij})$;
      $\mathcal{D}_i \leftarrow \mathcal{D}_i \cup \{(\mathbf{x}_0, i)\}$;
      $\mathcal{D}_{\text{syn}} \leftarrow \mathcal{D}_{\text{syn}} \cup \mathcal{D}_i$;
    **end**
**end**
**return** $\mathcal{D}_{\text{syn}}$

---

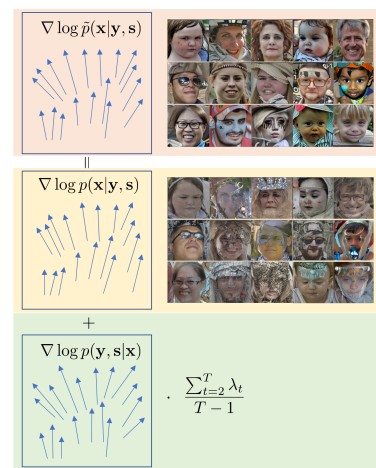

Figure 2: Qualitative comparison of face images generated by the adjusted score function $\nabla \log \tilde{p}(\mathbf{x}_t | \mathbf{y}, \mathbf{s})$ and the original score function $\nabla \log p(\mathbf{x}_t | \mathbf{y}, \mathbf{s})$.

Then, our ID-preserving sampling algorithm first draws a Gaussian sample $\mathbf{x}_T \sim \mathcal{N}(0, I)$. Afterwards, sequentially, the algorithm performs the following update for $t$ iterating from $T$ backwards to 1:

$$\mathbf{x}_{t-1} \leftarrow \mathbf{x}_t + \gamma \nabla \log \tilde{p}(\mathbf{x}_t | \mathbf{y}, \mathbf{s}) + \sqrt{2\gamma}\epsilon$$

where

$$\nabla \log \tilde{p}(\mathbf{x}_t | \mathbf{y}, \mathbf{s}) = \underbrace{\nabla \log p(\mathbf{x}_t | \mathbf{y}, \mathbf{s})}_{\text{original likelihood score}} + \frac{\sum_{t=2}^{T} \lambda_t}{T-1} \underbrace{\nabla \log p(\mathbf{y}, \mathbf{s} | \mathbf{x}_t)}_{\text{reversed likelihood score}} \tag{8}$$

Note that the original likelihood score in Eq. (8) can be evaluated by

$$\nabla \log p(\mathbf{x}_t | \mathbf{y}, \mathbf{s}) = \frac{\sqrt{\bar{\alpha}_t}}{\sqrt{1 - \bar{\alpha}_t}} \left( \hat{\mathbf{x}}_{\boldsymbol{\theta}}(\mathbf{x}_t, t, \mathbf{y}, \mathbf{s}) - \frac{\mathbf{x}_t}{\sqrt{\bar{\alpha}_t}} \right)$$

and the reversed likelihood score is given by a scaled inner product:

$$\nabla \log p(\mathbf{y}, \mathbf{s} | \mathbf{x}_t) = \kappa_{\mathbf{x}_t} \mathbf{y}^T \nabla f_{\boldsymbol{\phi}}(\mathbf{x}_t)$$

See Appendix B for the derivation of the above equations. As such, our ID-preserving sampling algorithm performs sampling by searching a trajectory in the vector field $\nabla \log \tilde{p}(\mathbf{x}_t | \mathbf{y}, \mathbf{s})$ that can maximize the adjusted conditional likelihood $\tilde{p}(\mathbf{x}_t | \mathbf{y}, \mathbf{s})$. See Algorithm 2 for the specific procedure.

**Remark.** Our proposed adjusted likelihood score differs from the original score by adding an extra scaled reversed likelihood score in Eq. (8). Consequently, as shown in Figure 2, the resulting vector field differs from the original vector field, which leads to different Langevin sampling trajectories and thus different sampling quality.

### 3.5 Synthetic Dataset Generation

In terms of the second question (**RQ2**): after training ID[3], with what sampling strategy is it possible to generate a synthetic face dataset on which SoTA face recognition models can be trained and perform well on challenging benchmarks?

Our proposed dataset-generating algorithm goes as follows: given $N$ target identities, we generate $N$ anchor embeddings distributed on the sphere: $\mathbf{w}_1, \mathbf{w}_2, ..., \mathbf{w}_N \in \mathbb{S}^{d-1}$ as uniformly as possible in the sense that each pair of the embeddings are maximally separated on the unit sphere[#]. For each anchor $\mathbf{w}_i$, we would like to generate $m$ identity embeddings perturbed around $\mathbf{w}_i$ while ensuring

---

[#]This is known as the Tammes problem (Tammes, 1930) for which there exists no exact solution for hypersphere $\mathbb{S}^{d-1}, d > 3$. However, one can use the optimization technique introduced in (Mettes et al., 2019).

that these $m$ identity embeddings get close to but different than $\mathbf{w}_i$. Specifically, to find these $m$ identity embeddings, we solve the following optimization problem $\mathbf{P}_i$:

$$
\min_{\mathbf{y}_{ij}, j=1,...,m} \left\| \begin{bmatrix} -\mathbf{w}_i^T- \end{bmatrix} \mathrm{norm} \left( \underbrace{\begin{bmatrix} | & | & \cdots & | \\ \mathbf{y}_{i1} & \mathbf{y}_{i2} & \cdots & \mathbf{y}_{im} \\ | & | & \cdots & | \end{bmatrix}}_{\mathbf{Y}_i} \right) - [\nu_{i1}, \nu_{i2}, ...\nu_{im}] \right\|_2^2 \tag{9}
$$

where the operator $\mathrm{norm}$ is column-wise normalization which normalizes each column of $\mathbf{Y}_i$ into a unit vector, and the desired similarity scores $\nu_{i1}, \nu_{i2}, ..., \nu_{im}$ are randomly generated from a continuous uniform distribution $\mathcal{U}[lb, ub]$. After solving Eq. (9), we are able to retrieve the $m$ optimal unnormalized vector $\mathbf{y}_{i1}^*, ..., \mathbf{y}_{im}^*$. These $m$ vectors are then normalized, yielding $m$ identity embeddings: $\mathrm{norm}(\mathbf{y}_{ij}^*)$, for $j = 1, ..., m$. Then, the resulting identity embeddings, along with face attributes, are fed into our generative models to generate face images. Finally, the entire dataset is generated by solving each $\mathbf{P}_i, i = 1, ..., N$, which yields $N$ identities, each with $m$ face images. The entire algorithm is summarized in Algorithm 3.

## 4    Experiments

In this section, we verify the effectiveness of ID$^3$ through empirical evaluation of the face dataset that ID$^3$ generates, and verify the performance of the SoTA face recognition model trained on this dataset in comparison with other baseline methods.

### 4.1    Dataset

**Training Dataset:**    We train our proposed ID$^3$ on FFHQ (Karras et al., 2019) dataset. The FFHQ (FaceForensics++) dataset is a large-scale dataset used for benchmarking and evaluating the performance of deep learning models in the field of face forensics. It is an extension of the original Face-Forensics dataset, which was designed to facilitate the development and comparison of methods for detecting and preventing face manipulation and deepfakes. In order to compare with DCFace (Kim et al., 2023), we also train ID$^3$ on CASIA-WebFace (Yi et al., 2014). The CASIA-WebFace dataset is used for face verification and face recognition tasks. This dataset contains 494,414 face images of 10,575 real identities collected from the web.

**Benchmarks:**    The performance of face recognition models is evaluated on various benchmark datasets: LFW (Huang et al., 2008), CFP-FP (Sengupta et al., 2016), CPLFW (Zheng and Deng, 2018), AgeDB (Moschoglou et al., 2017) and CALFW (Zheng et al., 2017). They are used to measure the impact of different factors on face image, such as pose changes and age variations.

### 4.2    Implementation Details

For our ID$^3$, we implement the denoising network with a U-net architecture and the projection module with a three-layer perceptron (hidden-layer size $(512, 256, 768)$) with ReLU activation. All

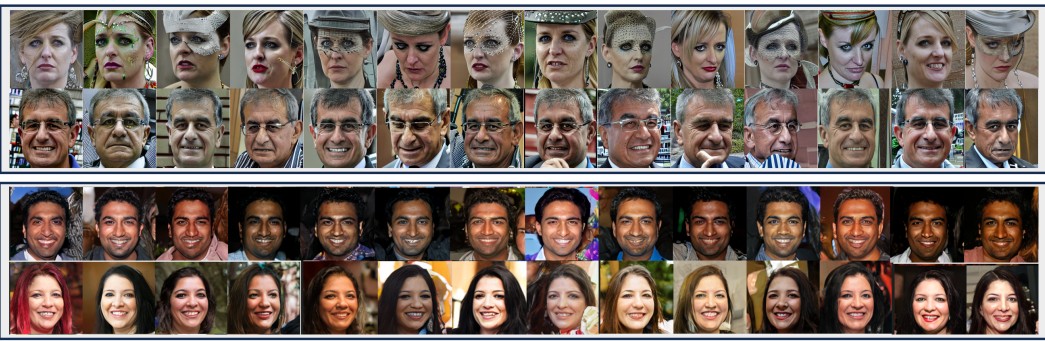

Figure 3: Uncurated samples generated by ID$^3$ (**Top**) and those by IDiff-Face (**Bottom**).

Table 1: SoTA Comparison. Face verification accuracy (%) of LResNet50-IR on different benchmarks when trained on synthetic datasets from ID$^3$ and state-of-the-art SFR generative models. For fairness, all methods generate face datasets of 10K identities each of which has 50 face images.

| Method | Training set | LFW | CFP-FP | CP-LFW | AgeDB | CA-LFW | Average |
|---|---|---|---|---|---|---|---|
| ID-Net | FFHQ | 84.83 | 70.43 | 67.35 | 63.58 | 71.50 | 71.53 |
| DigiFace | FFHQ | 88.07 | 70.99 | 66.73 | 60.92 | 69.23 | 71.19 |
| SFace | FFHQ | 91.43 | 73.10 | 73.42 | 69.87 | 76.92 | 76.95 |
| SynFace | FFHQ | 91.93 | 75.03 | 70.43 | 61.63 | 74.73 | 74.75 |
| IDiff-Face | FFHQ | 97.10 | 82.00 | 76.65 | 78.40 | 86.32 | 84.09 |
| **ID$^3$ (Ours)** | FFHQ | **97.28** | **85.00** | **77.13** | **83.78** | **89.30** | **86.50** |
| DCFace | FFHQ+CASIA | **98.55** | 85.33 | 82.62 | 89.7 | **91.6** | 89.56 |
| **ID$^3$ (Ours)** | CASIA | 97.68 | **86.84** | **82.77** | **91.00** | 90.73 | **89.80** |

models are implemented with PyTorch and trained from scratch using 8 NVIDIA Tesla V100 GPUs. Specifically, we set $\lambda_t \kappa_{\mathbf{x}_t} = 0.5 \cdot (1 - 1/(1 + \exp(-t/T))$ for the loss coefficients in Eq. (3), and use $T = 1,000$ for the diffusion model; training batch size is set to 16 and the total training steps $500,000$. We directly use a pre-trained face recognition (FR) model sourced from pSp (Richardson et al., 2021) as the identity feature extractor. Throughout the entire training process, these pre-trained models are frozen. In addition, we set # of identity embeddings $m = 25$ in Eq. (9) for each ID and match their embeddings with randomly selected attributes as conditioning signals for the diffusion model. For face recognition, we use LResNet50-IR (Deng et al., 2019), a variant of ResNet (He et al., 2016), as the backbone framework and follow the original configurations.

### 4.3 Performance Evaluation

We test the performance of the face recognition model trained on synthetic face data generated by ID$^3$ and compare against SoTA SFR generative models, including IDiff-Face (Boutros et al., 2023), ID-Net (Kolf et al., 2023), DigiFace (Bae et al., 2023), SFace (Boutros et al., 2022), SynFace (Qiu et al., 2021) and DCFace (Kim et al., 2023).

#### 4.3.1 Qualitative Results

Here, we illustrate a collection of face images generated by ID$^3$ as qualitative evaluation. Figure 3 shows the results for randomly sampled identities (IDs) under various attribute conditions; Obviously, when comparing different identities (inter-class), the essential intrinsic key information of each identity is still retained and can be easily identified. Also, different samples of each identity (intra-class) exhibit distinct diversity, stemming from variations in similarity scores ($\nu_{ij}$'s) and differences in face attributes as conditioning signals. In terms of the effect of our proposed adjusted score and the original score on the sampling algorithm, we observe that the face images generated by our proposed ID$^3$ exhibits much better quality and identity preservation than those generated by the original score function, as shown in Figure 2.

#### 4.3.2 Quantitative Results

We compare the accuracies of FR models trained on the synthetic face datasets generated by different generative models and demonstrate the results in Table 1.

As shown in Table 1, ID$^3$ demonstrates consistent superior performance, achieving the highest average accuracy of $86.50\%$, and outperforms other baselines in all benchmarks, notably scoring $83.78\%$ in AgeDB and $85.00\%$ in CFP-FP. This demonstrates the effectiveness of ID$^3$ in gaining pose and age control. Other methods, while effective to varying degrees, attain average scores below $86.50\%$ and are inferior to ID$^3$.

It is worth mentioning that ID$^3$, apart from using real data during training, does not introduce any real images as auxiliary data during the sampling phase. The synthetic data is directly used in the training of the face recognition model without undergoing any secondary or manual filtering. Additionally, when training the face recognition model using the synthetic data, no real images are introduced as auxiliary data. On the other hand, DCFace, as described and reported in (Kim et al., 2023), introduces real face images as auxiliary data during the training phase for face recognition.

Table 2: Ablation Study. Face verification accuracy (%) of LResNet50-IR when trained on synthetic datasets from $ID^3$ and other model variants. $ID^3$-$[lb, ub]$ represents an $ID^3$ variant using $lb$ and $ub$ as lower- and upper-bound for sampling $\nu_{ij}$'s. $ID^3$-random denotes a model variant that randomly sets anchors on the unit hypersphere for sample generation. $ID^3$-w/o-attribute denotes one that does not use attributes as conditioning signals. $ID^3$-w/o-reversed denotes one that removes the reversed likelihood score from Eq. (8) in the proposed ID-preserving sampling algorithm.

| Method | LFW | CFP-FP | CP-LFW | AgeDB | CA-LFW | Average |
|---|---|---|---|---|---|---|
| $ID^3$-w/o-reversed | 78.82 | 62.68 | 61.83 | 58.20 | 63.71 | 65.05 |
| $ID^3$-w/o-attribute | 97.12 | 85.57 | 81.70 | 87.50 | 89.48 | 88.27 |
| $ID^3$-$[0.7, 0.9]$ | 97.28 | 84.26 | 81.48 | 86.25 | 89.63 | 87.78 |
| $ID^3$-$[0.5, 0.7]$ | 97.38 | 85.00 | 81.10 | 86.63 | 90.13 | 88.05 |
| $ID^3$-random | 96.00 | 80.81 | 78.05 | 85.53 | 87.57 | 85.59 |
| **$ID^3$ (Ours)** | **97.68** | **86.84** | **82.77** | **91.00** | **90.37** | **89.80** |

This helps enhance the diversity of the training data and leads to slightly better results than $ID^3$ in the two benchmarks.

**Ablation study.** We further investigate the impact of each contributing component of $ID^3$ in generating a synthetic face dataset on SFR. This includes three ablation studies shown in Table 2: the effect of the reversed likelihood score in Eq. (8) on ID-preserving sampling algorithm ($ID^3$ vs. $ID^3$-w/o-reversed), the effect of using anchors in $ID^3$ ($ID^3$ vs. $ID^3$-random), and the effect of lower- and upper-bound of Uniform distribution for sampling $\nu_{ij}$'s.

In the first study, we compare $ID^3$ with $ID^3$-w/o-reversed, which removes the reversed likelihood score from Eq. (8) in the proposed ID-preserving sampling algorithm. We observed $ID^3$ consistently outperforms $ID^3$-w/o-reversed with large margins. This suggests the necessity of the inner-product term in the proposed loss function Eq. (3) and the reversed likelihood score in the adjusted likelihood score Eq. (8).

In the second study (cf. Appendix E), an appropriate smaller value of $lb$, if not exceeding a certain range, can increase the intra-class diversity, resulting in more diverse intra-class face images. This aligns with our objective of increasing intra-class diversity in the generated data to enhance the effectiveness of SFR. As per the constraints of Eq. (9), each generated identity embedding $\mathbf{y}_{ij}$ maintains the same identity as the anchor $\mathbf{w}_i$. This, along with our proposed inner-product term in Eq. (3), ensures consistent intra-class identities while introducing a significant amount of diversity.

In the third study, we demonstrate how effective it is to use maximally-separated anchors in $ID^3$ as compared to $ID^3$-random that randomly sets anchors on the unit hypersphere for sample generation. Clearly, $ID^3$-random does not yield as good results as $ID^3$. This is because the random sampling method only introduces one identity signal per ID, while the model requires a combination of attributes and identity signals. Attribute signals can only control explicit attributes, whereas identity signals control implicit properties. Introducing only one identity signal per identity implies insufficient intra-class diversity. Additionally, $ID^3$-random fails to regularize the relationship among different identities, leading to inadequate diversity among classes or aliasing issues with different identity signals. The identity signals obtained using $ID^3$ resolves the problem of aliasing between identity signals across classes, effectively improving intra-/inter-class diversity.

## 5 Conclusion

We have proposed $ID^3$, an identity-preserving-yet-diversified diffusion generative model for SFR. Our theoretical analysis regarding the training of $ID^3$ induces a new ID-preserving sampling algorithm and further, a dataset-generating algorithm that generates identity-preserving face images with inter-/intra-class diversity. Extensive experiments show that $ID^3$ outperforms existing methods in challenging multiple benchmarks.

**Limitations.** While $ID^3$, designed for the sake of privacy protection, achieves SoTA performance in SFR, there remains clear margins as compared to the FR performance when training with real-world face datasets such as MS1M. This suggests that the fake face dataset generated by $ID^3$ does not fully approximate the real-world faces. Future work might include closing this gap.

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

# ID³: Identity-Preserving-yet-Diversified Diffusion Models for Synthetic Face Recognition

## Supplementary Material

## A    Appendix A

We first present a lemma (Lemma A.1) that will be further used to prove Theorem 3.1.

**Lemma A.1.** *Given any two conditional probability density functions, $p(a|b)$ and $p(b|a)$, and a positive scalar $w > 0$, there exists another conditional probability density function $\tilde{p}$ such that*

$$\tilde{p}(a|b) = \frac{1}{Z_{b,w}} p(a|b)p(b|a)^w, \text{where } Z_{b,w} = \int p(a|b)p(b|a)^w da. \tag{A.1}$$

*Proof.* To show this result is equivalent to showing $\tilde{p}(a|b) \propto p(a|b) \cdot p(b|a)^w, w > 0$, which is equivalent to showing that, for any $b = b_0$,

$$\tilde{p}(a|b = b_0) \propto p(a|b = b_0) \cdot p(b = b_0|a)^w, w > 0 \tag{A.2}$$

Note that $\tilde{p}(a|b = b_0)$ is a function of $a$, i.e. there exists a function $f_{b_0,w}$ such that $p(b = b_0|a)^w := f_{b_0,w}(a)$. Let

$$Z_{b_0,w} := \int p(a|b = b_0)f_{b_0,w}(a)da \tag{A.3}$$

Then,

$$\frac{p(a|b = b_0) \cdot p(b = b_0|a)^w}{Z_{b_0,w}} = \frac{p(a|b = b_0) \cdot f_{b_0,w}(a)}{Z_{b_0,w}} \tag{A.4}$$

is a proper probability density function of $a$ with $b = b_0$ given. Therefore, Eq. (A.4) can be written as $\tilde{p}(a|b = b_0)$. The above proof holds true for any $b_0$, which concludes the proof for $\tilde{p}(a|b) \propto p(a|b) \cdot p(b|a)^w, w > 0$. Note that this result can be trivially extended to multivariate random variables.

$\square$

**Theorem 3.1.** *Minimizing $\mathcal{L}$ with regard to $\boldsymbol{\theta}$ is equivalent to minimizing the upper bound of an adjusted conditional data negative log-likelihood $-\log \tilde{p}(\mathbf{x}|\mathbf{y}, \mathbf{s})$, i.e.:*

$$\min_{\boldsymbol{\theta}} \mathcal{L}_{\boldsymbol{\theta}}(\mathbf{x}_0, \mathbf{y}, \mathbf{s}) \geq -\log \tilde{p}(\mathbf{x}|\mathbf{y}, \mathbf{s}) \tag{A.5}$$

*where*

$$\tilde{p}(\mathbf{x}|\mathbf{y}, \mathbf{s}) \propto p(\mathbf{x}|\mathbf{y}, \mathbf{s}) \cdot p(\mathbf{y}, \mathbf{s}|\mathbf{x})^{\frac{\sum_{t=2}^{T} \lambda_t}{T-1}} \tag{A.6}$$

*and $\mathbf{x}_0$ and $\mathbf{x}$ both refer to a raw image interchangeably.*

*Proof.* Recall that

$$\mathcal{L}_{\boldsymbol{\theta}}(\mathbf{x}_0, \mathbf{y}, \mathbf{s}) = \mathbb{E}_{t \sim \mathcal{U}[2,T]} \left[ \underbrace{\mu_t \left\| \mathbf{x}_0 - \hat{\mathbf{x}}_0^{(t)} \right\|_2^2}_{\text{denoising term}} - \underbrace{\lambda_t \kappa_{\mathbf{x}} \mathbf{y}^T f_{\boldsymbol{\phi}}\left(\hat{\mathbf{x}}_0^{(t)}\right)}_{\text{inner-product term}} \right] + \mathbb{E}_{q(\mathbf{x}_1|\mathbf{x}_0)} \left[ \underbrace{\frac{1}{2} \left\| \mathbf{x}_0 - \hat{\mathbf{x}}_0^{(1)} \right\|_2^2}_{\text{one-step reconstruction term}} \right] + C$$

$$= \mathbb{E}_{t \sim \mathcal{U}[2,T]} \left[ \underbrace{\mu_t \left\| \mathbf{x}_0 - \hat{\mathbf{x}}_0^{(t)} \right\|_2^2}_{\text{denoising term}} \right] + \mathbb{E}_{q(\mathbf{x}_1|\mathbf{x}_0)} \left[ \underbrace{\frac{1}{2} \left\| \mathbf{x}_0 - \hat{\mathbf{x}}_0^{(1)} \right\|_2^2}_{\text{one-step reconstruction term}} \right] + C - \mathbb{E}_{t \sim \mathcal{U}[2,T]} \left[ \underbrace{\lambda_t \kappa_{\mathbf{x}} \mathbf{y}^T f_{\boldsymbol{\phi}}\left(\hat{\mathbf{x}}_0^{(t)}\right)}_{\text{inner-product term}} \right]$$

$$\tag{A.7}$$

It can be shown that the reversed likelihood $p(\mathbf{y}, \mathbf{s}|\mathbf{x})$ is a joint vMF density (Xu et al., 2023; Hasnat et al., 2017):

$$p(\mathbf{y}, \mathbf{s}|\mathbf{x}) \overset{(1)}{=} p(\mathbf{y}|\mathbf{s}, \mathbf{x})p(\mathbf{s}|\mathbf{x}) \overset{(2)}{=} p(\mathbf{y}|\mathbf{x})p(\mathbf{s}|\mathbf{x}) \overset{(3)}{=} J_{\kappa_{\mathbf{x}}}^2 \exp\left(\kappa_{\mathbf{x}}(\mathbf{y}^T f_{\boldsymbol{\phi}}(\mathbf{x}) + \mathbf{s}^T F_a(\mathbf{x}))\right) \tag{A.8}$$

where $J_{\kappa_{\mathbf{x}}}$ is the normalizing constant and $F_a$ is the pretrained attribute predictor. Note that Equality (1) is obtained by the product rule of probability; Equality (2) is obtained by observing that $\mathbf{y}$ and $\mathbf{s}$ are conditionally independent when $\mathbf{x}$ is given; and Equality (3) is obtained by assuming that

$$p(\mathbf{y}|\mathbf{x}) = J_{\kappa_{\mathbf{x}}} \exp\left(\kappa_{\mathbf{x}} \cdot \mathbf{y}^T f_{\boldsymbol{\phi}}(\mathbf{x})\right), \quad p(\mathbf{s}|\mathbf{x}) = J_{\kappa_{\mathbf{x}}} \exp\left(\kappa_{\mathbf{x}} \cdot \mathbf{s}^T F_a(\mathbf{x})\right) \tag{A.9}$$

Note that these reasonable assumptions are also held in (Xu et al., 2023; Li et al., 2021; Hasnat et al., 2017). Now we can specify the value of the scalar $C$:

$$C = -\mathbb{E}_{t \sim \mathcal{U}[2,T]}\left[\lambda_t\left(\log J_{\kappa_{\mathbf{x}}}^2 + \mathbf{s}^T F_a(\mathbf{x})\right)\right] - \frac{n}{2}\log(2\pi) + D_{\mathrm{KL}}\left(q(\mathbf{x}_T|\mathbf{x}_0)||p(\mathbf{x}_T)\right) + \log Z \tag{A.10}$$

where $n = 3HW$ is the dimensionality of $\mathbf{x}$, and

$$Z = \int_{\mathbf{x}} p(\mathbf{x}|\mathbf{y},\mathbf{s}) \cdot p(\mathbf{y},\mathbf{s}|\mathbf{x})^{\frac{\sum_{t=2}^T \lambda_t}{T-1}} d\mathbf{x} = Z\left(\mathbf{y}, \mathbf{s}, \frac{\sum_{t=2}^T \lambda_t}{T-1}\right) \tag{A.11}$$

Note that $Z$ only depends on $\mathbf{y}$, $\mathbf{s}$ and $\frac{\sum_{t=2}^T \lambda_t}{T-1}$ and hence $C$ is a scalar that does not depend on the learnable parameter $\boldsymbol{\theta}$. Therefore, $\mathcal{L}$ can be rewritten into a sum of two parts: $\mathcal{L} = \mathcal{L}_1 + \mathcal{L}_2$, where

$$\mathcal{L}_1 = \mathbb{E}_{t \sim \mathcal{U}[2,T]}\left[\underbrace{\mu_t\left\|\mathbf{x}_0 - \hat{\mathbf{x}}_0^{(t)}\right\|_2^2}_{\text{denoising term}}\right] + \mathbb{E}_{q(\mathbf{x}_1|\mathbf{x}_0)}\left[\underbrace{\frac{1}{2}\left\|\mathbf{x}_0 - \hat{\mathbf{x}}_0^{(1)}\right\|_2^2}_{\text{one-step reconstruction term}}\right] - \frac{n}{2}\log(2\pi) + D_{\mathrm{KL}}\left(q(\mathbf{x}_T|\mathbf{x}_0)||p(\mathbf{x}_T)\right) \tag{A.12}$$

and

$$\mathcal{L}_2 = -\mathbb{E}_{t \sim \mathcal{U}[2,T]}\left[\lambda_t \log J_{\kappa_{\mathbf{x}}}^2 + \lambda_t \kappa_{\mathbf{x}} \mathbf{s}^T F_a(\mathbf{x}) + \underbrace{\lambda_t \kappa_{\mathbf{x}} \mathbf{y}^T f_{\boldsymbol{\phi}}\left(\hat{\mathbf{x}}_0^{(t)}\right)}_{\text{inner-product term}}\right] + \log Z \tag{A.13}$$

We recognize $-\mathcal{L}_1$ is the evidence lower bound (ELBO) of $\log p(\mathbf{x}|\mathbf{y},\mathbf{s})$ (Luo, 2022), i.e.

$$\mathcal{L}_1 \geq -\log p(\mathbf{x}|\mathbf{y},\mathbf{s}) \tag{A.14}$$

As for $\mathcal{L}_2$:

$$\mathcal{L}_2 = -\mathbb{E}_{t \sim \mathcal{U}[2,T]}\left[\lambda_t\left(\log J_{\kappa_{\mathbf{x}}}^2 + \underbrace{\kappa_{\mathbf{x}} \mathbf{y}^T f_{\boldsymbol{\phi}}\left(\hat{\mathbf{x}}_0^{(t)}\right)}_{\text{inner-product term}} + \kappa_{\mathbf{x}} \mathbf{s}^T F_a(\mathbf{x})\right)\right] + \log Z$$

$$= -\frac{1}{T-1}\sum_{t=2}^T\left[\lambda_t\left(\log J_{\kappa_{\mathbf{x}}}^2 + \underbrace{\kappa_{\mathbf{x}} \mathbf{y}^T f_{\boldsymbol{\phi}}\left(\hat{\mathbf{x}}_0^{(t)}\right)}_{\text{inner-product term}} + \kappa_{\mathbf{x}} \mathbf{s}^T F_a(\mathbf{x})\right)\right] + \log Z \tag{A.15}$$

Here we assume access to a perfect denoising module such that $\hat{\mathbf{x}}_0^{(t)} = \mathbf{x}_0 = \mathbf{x}$, for all $t$'s. Hence, $\mathcal{L}_2$ can be written as

$$\mathcal{L}_2 = -\frac{1}{T-1}\sum_{t=2}^T\left[\lambda_t\left(\log J_{\kappa_{\mathbf{x}}}^2 + \kappa_{\mathbf{x}} \mathbf{y}^T f_{\boldsymbol{\phi}}(\mathbf{x}) + \kappa_{\mathbf{x}} \mathbf{s}^T F_a(\mathbf{x})\right)\right] + \log Z \tag{A.16a}$$

$$= -\frac{1}{T-1}\sum_{t=2}^T\left[\lambda_t \log p(\mathbf{y},\mathbf{s}|\mathbf{x})\right] + \log Z \tag{A.16b}$$

$$= -\frac{1}{T-1}\left(\sum_{t=2}^T \lambda_t\right) \cdot \log p(\mathbf{y},\mathbf{s}|\mathbf{x}) + \log Z \tag{A.16c}$$

$$= -\log p(\mathbf{y},\mathbf{s}|\mathbf{x})^{\frac{\sum_{t=2}^T \lambda_t}{T-1}} + \log Z \tag{A.16d}$$

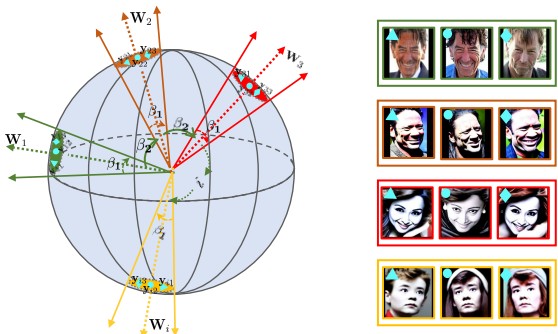

Figure A.1: An illustration of the dataset-generating algorithm.

Then,

$$\mathcal{L} = \mathcal{L}_1 + \mathcal{L}_2 \tag{A.17a}$$

$$\geq - \left[ \log p(\mathbf{x}|\mathbf{y},\mathbf{s}) + \log p(\mathbf{y},\mathbf{s}|\mathbf{x})^{\frac{\sum_{t=2}^{T} \lambda_t}{T-1}} \right] + \log Z \tag{A.17b}$$

$$= - \log \left( Z \cdot \tilde{p}(\mathbf{x}|\mathbf{y},\mathbf{s}) \right) + \log Z \tag{A.17c}$$

$$= - \log \tilde{p}(\mathbf{x}|\mathbf{y},\mathbf{s}) \tag{A.17d}$$

where Equality (A.17c) is obtained by applying Lemma A.1. This completes the proof. □

## B   Appendix B

In this section, we show the derivations of the following equations:

$$\nabla \log p(\mathbf{x}_t|\mathbf{y},\mathbf{s}) = \frac{\sqrt{\bar{\alpha}_t}}{\sqrt{1-\bar{\alpha}_t}} \left( \hat{\mathbf{x}}_{\boldsymbol{\theta}}(\mathbf{x}_t, t, \mathbf{y}, \mathbf{s}) - \frac{\mathbf{x}_t}{\sqrt{\bar{\alpha}_t}} \right) \tag{A.18}$$

$$\nabla \log p(\mathbf{y},\mathbf{s}|\mathbf{x}_t) = \kappa_{\mathbf{x}_t} \mathbf{y}^T \nabla f_{\boldsymbol{\phi}}(\mathbf{x}_t) + \kappa'_{\mathbf{x}_t} \mathbf{s}^T \nabla F_{\boldsymbol{a}}(\mathbf{x}_t) \tag{A.19}$$

Recall that in the main text, we showed that the adjusted likelihood score is a summation of the original likelihood score and the scaled reversed likelihood score:

$$\underbrace{\nabla \log \tilde{p}(\mathbf{x}_t|\mathbf{y},\mathbf{s})}_{\text{adjusted likelihood score}} = \underbrace{\nabla \log p(\mathbf{x}_t|\mathbf{y},\mathbf{s})}_{\text{original likelihood score}} + \frac{\sum_{t=2}^{T} \lambda_t}{T-1} \underbrace{\nabla \log p(\mathbf{y},\mathbf{s}|\mathbf{x}_t)}_{\text{reversed likelihood score}} \tag{A.20}$$

For the original likelihood score, we note that our proposed ID$^3$ itself is a conditional diffusion model. By virtue of the relation between the score and the denoising module (i.e. the Tweedie's Formula) in diffusion models (cf. Equation (133) in (Luo, 2022)), we are able to show that

$$\nabla \log p(\mathbf{x}_t|\mathbf{y},\mathbf{s}) = \frac{\sqrt{\bar{\alpha}_t}}{\sqrt{1-\bar{\alpha}_t}} \left( \mathbf{x}_0 - \frac{\mathbf{x}_t}{\sqrt{\bar{\alpha}_t}} \right) \tag{A.21}$$

Here, $\mathbf{x}_0$ can be approximated by denoising $\mathbf{x}_t$ via the trained denoising module

$$\mathbf{x}_0 \approx \hat{\mathbf{x}}_{\boldsymbol{\theta}}(\mathbf{x}_t, t, \mathbf{y}, \mathbf{s}) \tag{A.22}$$

## C   An Illustration of the Dataset Generating Algorithm

We illustrate the dataset generating algorithm in Figure A.1. First, $N$ anchor embeddings are generated on the sphere as uniformly as possible. Then, for each anchor, $m$ identity embeddings are generated around the anchor. This strategy ensures inter-class diversity while intra-class identity preservation is guaranteed. Colors show the correspondence between the generation procedure on the left and the generated samples on the right.

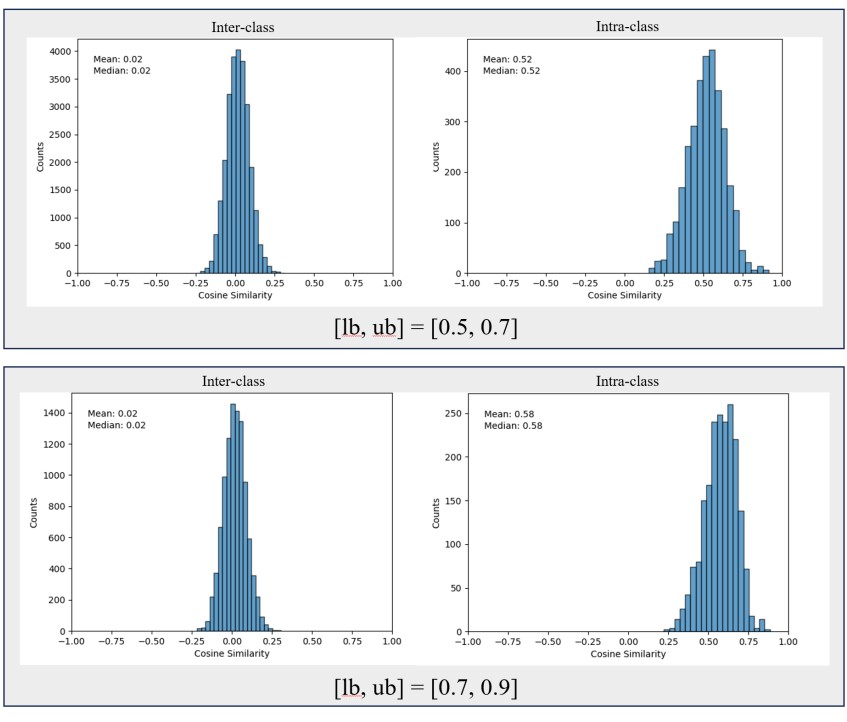

Figure A.2: The distribution of inter-class and intra-class similarities.

## D  Related Work

**Face Recognition.**    Face Recognition (FR) is the task of matching query imagery to an enrolled identity database. SoTA FR models are trained using margin-based softmax losses (Wang et al., 2018; Deng et al., 2019) on large-scale web-crawled datasets (Guo et al., 2016; Zhu et al., 2021). These datasets encompasses three characteristics in common (as mentioned in the introduction): (i) sufficient inter-class diversity; (ii) intra-class diversity; (iii) intra-class identity preservation. However, due to the introduction of various regulations restricting the use of authentic face data, researchers switch their attention to synthetic face recognition (SFR). We argue that the crux of SFR is to generate a training dataset that inherits the three characteristics above.

**GAN-based SFR models.** Most of the deep generative models for synthetic faces generation are based on GANs. DigiFace (Bae et al., 2023) utilizes a digital rendering pipeline to generate synthetic images based on a learned model of facial geometry and attributes. SFace (Boutros et al., 2022) and ID-Net (Kolf et al., 2023) train a StyleGAN-ADA (Karras et al., 2020) under a class-conditional setting. SynFace (Qiu et al., 2021) extends DiscoFaceGAN (Deng et al., 2020) using synthetic identity mix-up to enhance the intra-class diversity. However, the reported results shown by these models show significant performance degradation in comparison to FR trained on real data. This performance gap is mainly due to inter-class discrimination and small intra-class diversity in their generated synthetic training datasets. **Diffusion models for SFR.** Recently, Diffusion Models (DMs) (Ho et al., 2020; Lin et al., 2018; Song et al., 2020) gained attention for both research and industry due to their potential to rival GANs on image synthesis, as they are easier to train without stability issues, and stem from a solid theoretical foundation. Among SFR diffusion models, IDiff-Face (Boutros et al., 2023) achieves SoTA performance. On the basis of a diffusion model, it incorporates Contextual Partial Dropout to generate diverse intra-class images. However, IDiff-Face fails to regularize the relationship among different identities.

**Latent Diffusion Models.** There exist many diffusion-based models (LDMs) (e.g., Face0 (Valevski et al., 2023), PhotoMaker (Li et al.), FaceStudio (Yan et al., 2023), InstantID (Wang et al., 2024)) which use ID attributes to assist in generating images. However, we note that these LDMs are designed for image generation but not for SFR. Our empirical findings further suggests these LDMs do not perform reasonably well even when applied to SFR. We found that although these LDMs claim to be ID-preserving in the pixel space, their feature embeddings are not discriminative enough

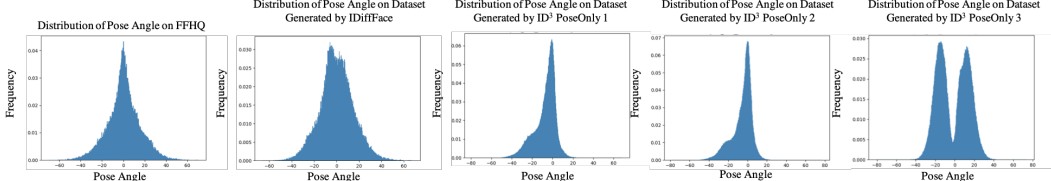

Figure A.3: Pose distribution of datasets generated by different models.

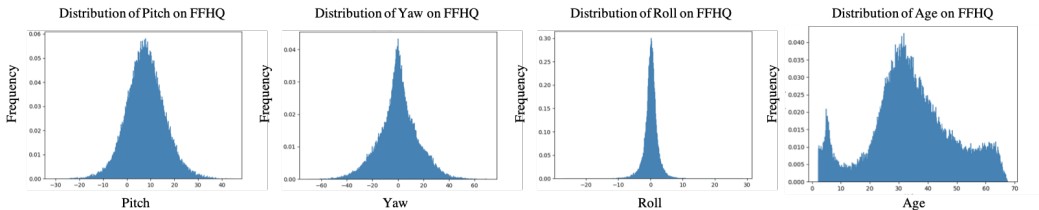

Figure A.4: Pose distribution (pitch, yaw, roll) and age distribution on FFHQ

for face recognition, since there is no inductive bias (neither loss functions nor architectures) to achieve face discriminativeness.

## E    Ablation Study (ii)

We show the inter-class and intra-class similarity in Figure A.2 when using $[0.5, 0.7]$ and $[0.7, 0.9]$ as the lower- and upper-bound $[lb, ub]$ for sampling $\nu_{ij}$'s in our proposed dataset-generating algorithm.

## F    Attribute Analysis of Generated Datasets

In this section, we perform an attribute analysis of the generated face datasets by ID$^3$. As shown in Figure A.3 and Figure A.4, from the distribution, we observe that the highest age in our training set is 70; the largest pose is 60 in degree and the smallest -60 in degree. Our model can interpolate within these ranges but is less likely to extrapolate outside of these ranges. To generate face images of large pose and high age, one can collect more such data and add them to the training dataset, which increases their occurrences during training.

To examine whether the attributes we used in our paper are fit for SFR tasks, we perform the following ablation study: as we increase intra-class pose variation, the SFR performances on cross-pose test sets (including CFPFP and CPLFW) are boosted whereas the performances on cross-age test sets (including AgeDB and CALFW) remain almost unchanged. Results and distribution plots are shown in Table A.1, Figure A.3 and Figure A.4. From these results, we observe that the distribution of pose angle on FFHQ, Dataset by IDiffFace, Dataset by ID PoseOnly 1 and 2 are all unimodal. And their performances are inferior to Dataset by ID PoseOnly 3 which exhibits multimodal distribution of pose angle.

Table A.1: Datasets generated by different models (Column 1), the attribute statistics for each dataset (Column 2, 3), and the FR performance of FR models trained on them, respectively (Column 4-8).

|  | Pose Mean | Pose Var | LFW | CFP-FP | CPLFW | AgeDB | CALFW |
|---|---|---|---|---|---|---|---|
| FFHQ | 11.44 | 233.97 | — | — | — | — | — |
| IDiffFace | 11.62 | 222.62 | 97.10 | 82.00 | 76.65 | 78.40 | 86.32 |
| ID$^3$ PoseOnly 1 | 8.21 | 109.76 | 95.33 | 78.41 | 73.48 | 79.76 | 86.03 |
| ID$^3$ PoseOnly 2 | 9.02 | 110.18 | 95.58 | 80.91 | 73.60 | 79.45 | 85.93 |
| ID$^3$ PoseOnly 3 | 14.14 | 247.05 | 95.83 | 82.87 | 75.77 | 79.45 | 86.90 |

