# OpenReview forum: "$\text{ID}^3$: Identity-Preserving-yet-Diversified Diffusion Models for Synthetic Face Recognition"
_NeurIPS.cc/2024/Conference — NeurIPS 2024 poster_

### Official Review · Reviewer_enp6 · 2024-06-21

**Soundness:** 2
**Presentation:** 3
**Contribution:** 2
**Rating:** 5
**Confidence:** 4

**Summary:**

This paper presents a method called $\text{ID}^3$ for the task of synthetic face recognition. The authors highlight that the accuracy of face recognition using generated data still lags behind that of training directly on real face data. They propose optimizing the generation process from the perspectives of diversity and consistency.

**Strengths:**

- Clear explanation of formulas and algorithm flow.
- Achieved SOTA results compared to methods from the past two years.

**Weaknesses:**

- There has been extensive research on ID preserving, and recent models based on LDM (e.g., Face0, PhotoMaker, FaceStudio, InstantID) can also be used for synthetic face recognition. The paper lacks analysis and comparative experiments on these models.
- The Face Attribute Conditioning Signal includes age and pose (pose angle range: [-90°, 90°]). However, the visual results in the paper do not reflect these attributes. The variation in pose is minimal, and there is no demonstration of different levels of age (which you mentioned as [0-100]).
- The paper devotes too much space to mathematical derivations and lacks intuitive visual results. For example, using different attributes and ID information to guide the model could be visualized by showing how the various layers of the Unet perceive this information.

**Questions:**

- What is the resolution of the training and generated images?
- How long does the training process take using 8 NVIDIA Tesla V100 GPUs?
- What is the image generation speed during inference?
- How much GPU memory is required for inference?
- How do the ID information and attribute information affect the Unet in the network structure? Is it through cross-attention or along with the timestep information?

**Limitations:**

Using ID attributes to assist in the generation results is already common in diffusion-based tasks. This method is essentially a conditional guided generation, and its technical contribution is limited.

---

> ### Author Rebuttal · Authors · 2024-08-05
>
> We are glad that Reviewer enp6 finds our formulas and algorithm flow clear and that ID$^3$ has advantages over existing models over the past two years. Here we respond to your questions as follows. Hopefully it will address your concerns.
>
> **Response to Weakness 1**
>
> Thanks for pointing out these interesting works on LDMs. However, we would like to clarify that the primary goal of LDMs differs fundamentally from SFR. These LDMs aim to change styles or expressions given a base face image whereas SFR is to generate a fake face dataset **all from scratch** in place of the real one to achieve face recognition. Yet, we were curious about LDMs' performance on SFR as well before starting this research, although they were not designed for SFR. Among LDMs, we chose the best performing model InstantID and ran multiple tests on it. We empirically found that the best test result is no better than: 81.5 on LFW, 68.9 on CFP-FP, 62.9 on CP-LFW, 62.9 on AgeDB, 63.8 on CA-LFW, with an average of 68.01. These results are far behind the results of the SFR generative model family: SynFace, SFace, ID-Net, DigiFace, iDiffFace, DCFace and ID$^3$. (Please compare these results with those in Table 1.)
>
> We investigated the reasons: we found that while these LDMs claimed to be ID-preserving in the pixel space, their feature embeddings are not discriminative enough for face recognition, since there was no inductive bias (neither loss functions nor architectures) to achieve face discriminativeness.  Hence, there is no reason for them to work reasonably well in SFR. In our submitted work, we followed the prevailing practice in SFR by benchmarking ID$^3$ against SFR generative models other than these LDMs. Thanks for the suggestion, though. We will add these comparisons and discussions to the manuscript.
>
> **Response to Weakness 2**
>
> The pose angle range and the age range in Line 134 are for theoretical purpose and are the domains of definition only. The extreme values (e.g. -90 in degree, 90 in degree, 100 in age) have low occurrences even in the training dataset and thus has low prior probability to be captured from ID$^3$. We demonstrate the distribution of age and pose in the training set shown in Fig. 2 in the **one-page PDF**.
>
> From the distribution, we observe that the highest age in our training set is 70; the largest pose is 60 in degree and the smallest -60 in degree. Our model can interpolate within these ranges but is less likely to extrapolate outside of these ranges. To generate face images of large pose and high age, one can collect more such data and add them to the training dataset, which increases their occurrences during training.
>
> **Response to Weakness 3**
>
> Thanks for the suggestion. For intuitive understanding of our work, we will add more visual results to the manuscript which demonstrate how ID$^3$ works while generating different identities of different poses and ages. Now that the ID embeddings and face attributes affect UNet through the cross-attention mechanism, we visualize the attention map in the UNet given different identities and face attributes, as shown in the **one-page PDF**.
>
> From the visualization result, we observe that ID$^3$ first determines the general pose and the outline of a face, and then fill in the facial details and background noise. This provides an interesting insight for understanding how the diffusion model works in ID$^3$.
>
> Besides, we would like to divert your attention to Fig. 2 in the main paper, which gives insights for how our proposed ID-preserving sampling algorithm (Alg. 2) works. We observe that in the middle of Fig. 2, the original score function $\nabla \log p(\mathbf{x}_t | \mathbf{y}, s)$ generates low-quality face images with cluttered facial details whereas the adjusted score function $\nabla \log \tilde{p}(\mathbf{x}_t | \mathbf{y}, s)$ produces high-quality results of various attributes. One can find the correspondence between the adjustment of the vector field and the resulting images.
>
> **Response to Q1:**
> The resolution of the training and generated images is 64x64.
>
> **Response to Q2:**
> It takes roughly one week to train ID$^3$ when using 8 NVIDIA Tesla V100 GPUs.
>
> **Response to Q3:**
> The image generation speed is 0.5 images/second.
>
> **Response to Q4:**
> For inference, it takes 21G GPU memory.
>
> **Response to Q5:**
> It is through cross-attention. The ID information and attribute information both acts as keys and values in the cross-attention mechanism while $\mathbf{z}_t$ acts as queries.
>
> **Response to Limitation:**
>
> Yes, there indeed exist many diffusion-based models (LDMs) that use ID attributes to assist in generating images. But again they are designed for image generation but not for SFR. We argue that image generation and SFR are two different tasks with two different goals. Face image generation is to change styles or expressions given a base face image whereas SFR is to generate a fake face dataset **all from scratch** in place of the real one to achieve face recognition. Moreover, these LDMs do not perform reasonably well even when applied to SFR. It is worth noting that our technical contribution goes beyond image generation and identify three important characteristics for successful SFR: intra-class diversity, inter-class diversity and ID preservation. Various technical contributions are proposed in this paper to achieve these three characteristics, respectively. For example: ID-preserving loss, Theorem 3.1 and Algorithm 2 are proposed to achieve ID preservation; anchor generation (solving the Tammes problem) (see Line 191-193) is to achieve inter-class diversity; solving Eq. (9) is to achieve intra-class diversity.
>
> Therefore, our technical contribution consists of not only deriving a conditional diffusion model but also a new ID-preserving sampling algorithm as well as a face dataset generation algorithm that respects face distribution on the face manifold to achieve intra-class diversity, inter-class diversity and ID preservation.

---

> > ### Comment · Reviewer_enp6 · 2024-08-09
> >
> > The authors have partially addressed my concerns, and I will give them a higher score. If you can provide me with an anonymous code link to check details about how the Attribute Conditioning Signal and Identity Loss work, I can further increase the score.

---

> > > ### Author Response · Authors · 2024-08-12
> > >
> > > Sorry for the delayed response. We just have finished refactoring the entire project for better readibility, and here goes the anonymous link to the project: [https://anonymous.4open.science/r/id3_sfr-FB8B/](https://anonymous.4open.science/r/id3_sfr-FB8B/). To check the details about how the Attribute Conditioning Signal works, you may refer to the class *IdPoseAgeFeaEmbedder* defined in ./ldm/modules/encoders/modules.py; to check the details about how identity loss works, you may refer to the function *id_preserving_loss* defined in the class DDPM in ./ldm/models/diffusion/ddpm.py. Please let us know if you have any further questions or concerns.

---

### Official Review · Reviewer_37yP · 2024-06-30

**Soundness:** 2
**Presentation:** 3
**Contribution:** 2
**Rating:** 4
**Confidence:** 4

**Summary:**

This paper focuses on synthetic face recognition and proposes to concentrate on three aspects: inter-class diversity, intra-class diversity, and intra-class identity preservation. Based on those, an ID-preserving loss is employed to generate diverse but identity-preserving facial images. This paper also demonstrates the proposed loss is equal to lower bound of an adjusted conditional log-likelihood over ID-preserving data.

**Strengths:**

1. This work is well-written and well-organized. It brings some insights for SFR.
2. The idea of 3 aspects is good, and quite general for SFR
3. The proposed method shows advances when using the FFHQ dataset

**Weaknesses:**

Here are several concerns regarding this work:
1. The idea of Attribute Conditioning Signal is not fit for synthetic face recognition tasks, because factors contributing to solid FR training cannot be determined by simply adjusting face attributes. One reason is that the attribute network (e.g., pose, age) is not generalized enough, as the pre-trained models are obtained from relatively small-scale datasets compared to FR datasets. Additionally, the authors have not addressed which attributes are effective for FR, leaving this important question unanswered.

2. The performance trained on FFHQ dataset appears good; however FFHQ dataset has explicitly banned its use for face recognition applications. Furthermore, FFHQ is relatively small(210k images) which doesn’t contain enough diversity, that’s the reason facial attributes can be of improvement in this experiment. For more details on FFHQ please refer to: https://github.com/NVlabs/ffhq-dataset

3. When it comes to the relatively large dataset CASIA-WebFace, the improvement over DCFace is marginal. One problem is that DCFace is trained with CASIA-WebFace only, not the FFHQ+CASIA mentioned by the author.

4. Experiments are not sufficient. For example, DCFace provides experiment results on 3 data volumes: 500k, 1M and 1.2M. These are not included in this paper.

5. There are some typos, for example, Y_i should be given in line 194

**Questions:**

Based on Algorithm 3, the pipeline would be: firstly, generate multiple embeddings close to the anchor; Then use the diffusion model to synthesize the images. The questions are:
1. Given the unpack(Yi) and generate different attributes, the generated identity image would be affected by the attributes. Do the authors test whether the identities change given the different face attributes on the test? How to make sure the generated identity aligns with the input embedding in the generation phase?
2. How to make sure the diffusion model can generalize(recognize) this specific input embedding, considering the training embedding only covers a small range of the available embedding(training set)?

**Limitations:**

This paper shows an attempt to generate diverse facial images of each identity. However, what makes solid FR training is not studied. Furthermore, domain GAP exists when adopting the trained diffusion model for generation, the input embedding might differ from the embedding of the synthesized image. Consequently, the focus is on changing facial attributes but preserving identity is interesting but the overall improvement is marginal, and FFHQ has license issues related to the application of Face Recognition. I think it will be more convincing if the results of CASIA-WebFace under 1M and 1.2 M settings are presented.

---

> ### Author Rebuttal · Authors · 2024-08-05
>
> We are glad that Reviewer 37yP appreciates our work for its insight, generality and effectiveness. Here we respond to your questions as follows. Hopefully it will address your concerns.
>
> **Response to Weakness 1:**
>
> Thanks for pointing it out. Factors contributing to solid FR training are complicated, and it is impossible to disentangle them all from identity and to quantify them all explicitly and accurately. But it does not suggest that this problem is totally unsolvable. Some attributes can be easily quantified and disentangled from identity such as age and pose, while others intertwine with identity such as gender and micro-features. In this paper, for now we explicitly explore those quantifiable attributes (e.g. age and pose) by injecting them into ID$^3$ as conditioning signals and leave the unquantifiable attributes automatically and flexibly captured by the U-Net of ID$^3$ via model training. We find this practice leads to sufficiently good performance on our SFR task. Our proposed ID$^3$ serves as a framework into which more quantifiable attributes can be introduced, which we leave for future exploration. We will add more discussions to the manuscript.
>
> To examine whether the attributes we used in our paper are fit for SFR tasks, we perform the following ablation study: as we increase intra-class pose variation, the SFR performances on cross-pose test sets (including CFPFP and CPLFW) are boosted whereas the performances on cross-age test sets (including AgeDB and CALFW) remain almost unchanged. Results and distribution plots are shown in Tab. 1 and Fig. 1 in the **one-page PDF**. From these results, we observe that the distribution of pose angle on FFHQ, Dataset by IDiffFace, Dataset by ID$^3$ PoseOnly 1 and 2 are all unimodal. And their performances are inferior to Dataset by ID$^3$ PoseOnly 3 which exhibits multimodal distribution of pose angle. When comparing the results of ID$^3$ (Tab. 1 in the main text) and the results of ID$^3$ PoseOnly 3, we conclude that age is another essential attribute that should be injected into ID$^3$ as conditioning signals. We also would like to divert your attention to Table 2 in the main paper, which shows that without using attributes as conditioning signals, the performance is inferior to our original ID$^3$. These empirical findings suggests that the attributes as conditioning signals are fit for SFR tasks.
>
> Regarding the generalization of the attribute network, for simplicity and efficiency, the attribute network we used in our paper is a lightweight model, which we find performs efficiently and reasonably well in our task. Using a more powerful attribute predictor would further benefit our SFR task.
>
> **Response to Weakness 2:**
>
> The use of FFHQ as training data is for the sake of fairness, as SFR community all use FFHQ for model training. To have a fair comparison, we make sure in our paper that all competing models are trained on FFHQ and evaluated on standard benchmarks. Besides, we further use a larger dataset, CASIA, for training, as in Tab. 1.
>
> **Response to Weakness 3:**
>
> According to the original paper of DCFace (see the beginning of Section 5), the authors of DCFace claim that
>
> *"For $G_{id}$ which generates ID images, we adopt the publicly released unconditional DDPM [25] trained on FFHQ [36]. For $G_{mix}$, we train it on CASIA-WebFace [29] after initializing weights from $G_{id}$."*
>
> This suggests the final performance of DCFace depends on FFHQ and CASIA. Furthermore, we successfully replicated its performance reported in their paper using their official implementation: https://github.com/mk-minchul/dcface.git, which suggests DCFace is indeed trained on FFHQ+CASIA.
>
> Back to the argument that "the improvement over DCFace is marginal", note that ID$^3$ trained on CASIA only and DCFace is trained on FFHQ+CASIA, and yet ID$^3$ outperforms DCFace.
>
> **Response to Weakness 4:**
>
> Thanks for the suggestion. Here we show more results on the larger data volume with the base scale of 1.2M (>=):
> |Models|Add Real Faces To FR Training |CFP-FP | CPLFW | AgeDB | CALFW | average |
> |----------|----------|----------|----------|----------|----------|----------|
> |DCFace|Yes (15% Real Faces)|88.40| 84.22|90.45| 92.38|90.86|
> |ID$^3$|No (0% Real Faces)|89.89|84.98|91.15| 92.02|91.15|
>
> **Response to Question 1, 2:**
>
> Yes, we did run the relevant tests, as shown in Fig. A.2 in the supplement. Given different face attributes and an $\mathbf{Y}_i$, the intra-class cosine similarity ranges from 0.25 to 0.9 whereas the inter-class cosine similarity ranges from -0.25 to 0.25. This suggests that, given $\mathbf{Y}_i$, an identity do not change as we vary attributes only. For visualization, in Fig. 3 in the main paper, we also demonstrate the invariance of identity when attributes vary.
>
> The alignment between the generated identity and the input embedding stems from two factors: the manifold manifestation of the pretrained FR model $f_{\phi}$ and the generalizability of ID-preserving loss. First, the pretrained FR model $f_{\phi}$ is trained using ArcFace loss, which forces similar face embeddings to have large cosine values on manifold ($S^{d-1}$) and dissimilar face embeddings to have small values. Our proposed Alg. 3 (which solves Eq. (9) using $lb=0.7$ and $ub=0.9$) is based upon this manifold manifestation and ensures that $\mathbf{y}_i$ is similar to $\mathbf{w}_i$ with some inherent variations (intra-class diversity). Second, the ID-preserving loss maximizes the inner product between an input embedding and the embedding of the generated face image; we empirically find this loss function generalizes well to the embeddings that are close to (but not equal to) an anchor embedding on the manifold ($S^{d-1}$). We also experiment with $ub=1.0$, but it does not yield better SFR performance. We believe the choice of $lb$ and $ub$ is critical to this generalization of recognizing $\mathbf{y}_i$. We investigated this effect in Ablation (ii) (see Appendix E).

---

> > ### Comment · Reviewer_37yP · 2024-08-13
> >
> > Thank the authors for their rebuttal, which addresses part of my concerns. It impresses me in terms of directly using numerical facial embedding for SFR.
> >
> > However, I still have the following concerns:
> >
> > ---
> > **C1[minor]. Weakness 2:'SFR community all use FFHQ for model training':**
> >
> > I partly agree. I am aware that some of these SFR methods use pretrained generative models on FFHQ to provide Identity image (DCFace), and pre-mixing up facial image (SynFace). However, DigiFace does not involve training on FFHQ, and DCFace employs a pretrained model (on FFHQ) to provide style images while training a diffusion model on CASIA-WebFace. Given that the FFHQ dataset has restrictions concerning its application to FR, I think we should be careful when directly training a synthesizing model on FFHQ before the SFR community obtains permission to use this dataset.
> >
> > I agree that FFHQ can be adopted for reference, but I think it is better to present the SFR model trained on CASIA-WebFace at different data volumes.  Since CASIA-WebFace is a larger dataset, experiments conducted on it would be more convincing and general.
> >
> > ---
> > **C2[minor]. Weakness 4: 'Add Real Faces To FR Training'**
> >
> >  DCFace doesn't add real face sto FR training, instead, they add the synthesized image into the generated dataset 'we include the same $X_{id}$ for 5 additional times for each label' in Section 5.2. The $X_{id}$  is generated by the $G_{ID}$
> > , 'For sampling ID images, we generate 200, 000 facial images from $G_{ID}$' in Section 3.3. This means that the images used in the dataset are entirely synthetic. It’s unclear how '15%' is calculated in the rebuttal.
> >
> > ---
> > **C3[Major] . Weakness 4 : 'results of 1.2M'**
> >
> >  (1) I have recommended that the authors include experiments of 1M and 1.2M to make the paper more convincing in the original review opinion. However, only 1.2M results are given. Considering the limited rebuttal time it is understandable.
> >
> > (2) Normally results of 5 evaluation datasets are provided: LFW, CFP-FP, CPLFW, AgeDB, CALFW. The LFW results of 0.5M given by the manuscript are vastly worse than the competitor DCFace in Table 1. However, the LFW result is missing in the rebuttal, making the average of the rest marginally higher than DCFace. I recommend putting all the evaluation datasets especially **LFW** results for a fair comparison.
> >
> > I will keep my current score unless the full evaluation results are provided for a larger data volume.

---

> > > ### Author Response · Authors · 2024-08-14
> > >
> > > **Response to C1:**
> > >
> > > Thanks for the suggestion. Yes, we agree that care shall be taken when using FFHQ.
> > >
> > > ----------------------------------------------------------------------------------------------------------
> > > **Response to C2:**
> > >
> > > We apologize for the confusion caused by '15%'. The number '15%' was due to our false memory. We mistakenly cited this result from somewhere else. After having carefully examined the DCFace paper again, however, it suddenly occurred to us again that, to achieve promising results, DCFace requires manual data-filtering on the generated data. This can be seen from Sec 3.3 in the original paper of DCFace (https://arxiv.org/pdf/2304.07060):
> > >
> > > *"ID Image Sampling. For sampling ID images, we generate 200,000 facial images from $G_{id}$, from which we remove faces that are wearing sunglasses or too similar to the subjects in CASIA-WebFace with the Cosine Similarity threshold of 0.3 using $F_{eval}$. We are left with 105,446 images."*
> > >
> > > Moreover, in Style Image Sampling (see the 2nd paragraph of Sec. 3.3), the authors explored the manual sampling $X_{sty}$ from the pool of images whose gender/ethnicity matches that of $X_{id}$.
> > >
> > > We also have noted that the authors included *the same $X_{id}$ for 5 additional times for each label* and that *$X_{id}$ is generated by $G_{id}$ trained on FFHQ using DDPM* (see the beginning of Sec. 5 in the DCFace paper). This suggests that to generate ID embeddings, the authors made use of additional data, FFHQ, while using CASIA as training data. ID$^3$, on the other hand, uses CASIA only.
> > >
> > > In general, the authors of DCFace established several manual-filtering standards and used them to clean the data generated by their model. This practice is not in line with the standards of the SFR community. As we have claimed in our paper, in Line 40-41, *"Also note that, critically, the SFR dataset generation process should be fully automated without manual filtering or introducing auxiliary real face samples."*
> > >
> > > Furthermore, the manual-filtering used in DCFace is a time-consuming process. When scaling up to larger data volumes, DCFace is less likely to attain promising results whereas ID$^3$ is fully automated without any form of manual filtering and can linearly scale up to any data volume (please see our response to REVIEWER qiUe on Weakness 2 where we have analysed the time complexity of ID$^3$ which scales linearly with $m$ and $N$).
> > >
> > > The above were the points we would like to have made in the table. Here we clarify it as follows:
> > >
> > > |Models | Data |	Requires Manual Filtering	|LFW| CFP-FP|	CPLFW|	AgeDB	|CALFW	|average|
> > > |----------|----------|----------|----------|----------|----------|----------|----------|----------|
> > > |DCFace	| CASIA+FFHQ|Yes |98.83 |88.40	|84.22	|90.45|	92.38|	90.86|
> > > ID$^3$  |CASIA|    No |	97.73 |89.89	|84.98	|91.15	|92.02	|91.15 |
> > >
> > > ----------------------------------------------------------------------------------------------------------
> > >
> > > **Response to C3**
> > >
> > > Thanks for pointing it out. We would like to clarify that the average result has taken the LFW result into account already. We were unable to show the LFW result because of the character limit (6000) in the rebuttal. Here we show the full evaluation result. You may check whether the average value took LFW into account: 90.86 = (98.83 + 88.40 + 84.22 + 90.45 + 92.38) / 5; 91.15 = (97.73 + 89.89 + 84.98 + 91.15 + 92.02) / 5.
> > >
> > > |Models | Data |	Requires Manual Filtering	|LFW| CFP-FP|	CPLFW|	AgeDB	|CALFW	|average|
> > > |----------|----------|----------|----------|----------|----------|----------|----------|----------|
> > > |DCFace	| CASIA+FFHQ|Yes |98.83 |88.40	|84.22	|90.45|	92.38|	90.86|
> > > ID$^3$  |CASIA|    No |	97.73 |89.89	|84.98	|91.15	|92.02	|91.15 |
> > >
> > > From these results, we observe that ID$^3$ outperforms DCFace on three test sets out of five with a higher averaged performance (91.15>90.86). And considering that DCFace **requires manual-filtering to refine the generated data** and **uses additional data for generation** (see the response to C2), our proposed model ID$^3$ (operating as a unified automated system without any form of manual filtering and using CASIA only) is more promising and efficient, especially in generating larger and larger synthetic face datasets.
> > >
> > > On the theoretical side, ID$^3$ provides theoretical insights and advantages over existing works such as DCFace. Our work suggests that inductive biases regarding face manifold and distribution shall be introduced into an SFR generator as a whole.
> > >
> > > Therefore, we believe the existence of ID$^3$ will inspire more interesting works to appear in this community which further advances the field of SFR. Code will be made publicly available upon the revision of the manuscript.

---

> > > > ### Comment · Reviewer_37yP · 2024-08-14
> > > >
> > > > Thank the authors for the clarification.
> > > >
> > > > I can't agree with 'We were unable to show the LFW result because of the character limit (6000) in the rebuttal.' Why other numbers are not discarded? You can adjust the text to present the full results, additionally, it can also be provided in the pdf file.
> > > >
> > > > Furthermore, the results of IDiff-Face reported in this manuscript(average of 84.09) are significantly lower than those reported in the original IDiff-Face (average of 87.00, without data augmentation). The authors state that both methods used the FFHQ dataset, yet the results achieved by the proposed method (average of 86.50) are still worse than the originally reported IDiff-Face results (87.00). The difference between the original IDiffFace and the implemented IDiffFace should be pointed out in Table 1.
> > > >
> > > > Additionally, the results of training the model on the CAISA-WebFace dataset are of marginal improvement( 0.24% improvement on 0.5M and 0.29% on 1.2M).

---

> > > > > ### Author Response · Authors · 2024-08-14
> > > > >
> > > > > We thank REVIEWER 37yP for the response. Here we would like to further make a few clarifications:
> > > > >
> > > > > First, we were in a rush to submit the rebuttal in the very last minute and were unable to put all other results into the PDF file before the deadline. And we did not omit the LFW results on purpose. Moreover, the average values we reported in the first place already took into account the LFW results. Now that it has been clarified (see response to C3), we think this issue can be closed.
> > > > >
> > > > > Second, regarding the results of IDiff-Face, as we have clarified in the submitted paper, in Line 251, "The “IDiff-Face” results in Table 1 are obtained by using the pretrained model provided in its official repository https://github.com/fdbtrs/IDiff-Face." Using their pretrained model downloaded from the official IDiff-Face repository, we obtained the result of 84.09 rather than 87.00 as reported in the IDiff-Face paper. According to the research replication policy, we reported 84.09 in our submission and clarified it in Line 251. Thanks for the suggestion. We will point it out in Table 1 in the revised manuscript.
> > > > >
> > > > > Finally, we think it is ***unfair*** to say the improvements are marginal. As we mentioned above (see responses to C2 and C3), DCFace requires manual filtering to post-process the generated data and resorts to additional data, whereas ID$^3$ operates as a unified automated algorithm without any form of manual filtering and additional data. And yet, ID$^3$ can still yield SoTA or even better performance.

---

### Official Review · Reviewer_ThNX · 2024-07-09

**Soundness:** 3
**Presentation:** 3
**Contribution:** 3
**Rating:** 5
**Confidence:** 4

**Summary:**

This paper proposes ID3, an identity-preserving-yet-diversified diffusion model for generating synthetic face data for face recognition. ID3 leverages identity embeddings and facial attributes to control inter-class and intra-class diversity of generated faces while preserving intra-class identity consistency, demonstrating state-of-the-art performance on multiple synthetic face recognition benchmarks.

**Strengths:**

See questions section in detail.

**Weaknesses:**

See questions section in detail.

**Questions:**

The paper addresses a well-motivated and important problem in the field of synthetic face recognition. The proposed ID3 model demonstrates significant innovation in leveraging diffusion models conditioned on identity embeddings and facial attributes to generate diverse yet identity-consistent synthetic faces. Moreover, the theoretical analysis provided in the paper, which proves the equivalence between minimizing the proposed loss function and maximizing a lower bound on an adjusted data likelihood, lends credibility and rigor to the proposed approach.

However, there is room for improvement in the presentation and writing of the manuscript. One area that could benefit from further clarification is the explanation of notations and symbols used in the mathematical formulas. For instance, the meaning of the variable d in S
d−1 is not clearly defined, which may lead to confusion for readers. Additionally, the formatting and typesetting of some equations, such as Equation 3, could be enhanced to improve readability and aesthetic appeal.

**Limitations:**

The authors have adequately addressed the limitations.

---

> ### Author Rebuttal · Authors · 2024-08-05
>
> **Question: One area that could benefit from further clarification is the explanation of notations and symbols used in the mathematical formulas. Additionally, the formatting and typesetting of some equations, such as Equation 3, could be enhanced to improve readability and aesthetic appeal.**
>
> **Response:** We are glad that Reviewer ThNX finds our proposed model innovative. And thanks for pointing out these issues regarding the presentation and writing. The quantity $d$ in $S^{d-1}$ is the dimensionality of the face embedding space. We will clarify them in the manuscript and rewrite some equations (e.g. Equation 3) for better readability and aesthetic appeal.

---

> > ### Comment · Reviewer_ThNX · 2024-08-12
> >
> > Thank you for your detailed response. After careful consideration of the points raised in your rebuttal, I have decided to maintain my original rating for the paper.

---

### Official Review · Reviewer_qiUe · 2024-07-21

**Soundness:** 4
**Presentation:** 4
**Contribution:** 4
**Rating:** 8
**Confidence:** 5

**Summary:**

The paper "ID3: Identity-Preserving-yet-Diversified Diffusion Models for Synthetic Face Recognition" introduces a novel synthetic face recognition (SFR) approach using diffusion models. It focuses on maintaining identity consistency while providing high diversity in generated face images. The proposed ID3 model leverages identity-preserving losses and a structured sampling algorithm that respects identity characteristics. This effectively addresses the common pitfalls of existing SFR approaches that lead to poor generalization on real-world data.

**Strengths:**

*   **Originality**: The paper presents an innovative use of diffusion models tailored to synthetic face recognition, emphasizing identity preservation.
*   **Quality**: Demonstrated improvement over state-of-the-art models through extensive benchmarking.
*   **Clarity**: Exceptionally clear presentation and thorough explanation of the methodology and results.
*   **Significance**: This paper addresses significant challenges in synthetic data generation and offers substantial benefits for training more robust and generalizable face recognition systems.

**Weaknesses:**

*   **Generalization**: Additional tests on further diversified real-world datasets could strengthen the generalization claims.
*   **Complexity**: It would be beneficial to have details on the computational demands and scalability of the model when deployed in practical, real-world scenarios.

**Questions:**

1.  What measures have been taken to ensure the model's robustness against diverse ethnic and age groups, given the model's reliance on identity embeddings?
2.  Are there potential improvements or variations in the diffusion model that could further enhance identity preservation without sacrificing diversity?
3.  How does the model perform under constrained computational resources, and are there any strategies for optimizing its efficiency?

**Limitations:**

The paper discusses potential limitations, including the need for extensive computational resources and the model's performance dependency on the quality of input identity embeddings. It also mentions the ongoing challenge of bridging the gap between synthetic and real-world face recognition performance.

---

> ### Author Rebuttal · Authors · 2024-08-05
>
> We are glad that Reviewer qiUe appreciates our work in terms of originality, quality, clarity and significance. Here we respond to your questions as follows. Hopefully it will address your concerns.
>
> **Weakness 1: Additional tests on further diversified real-world datasets could strengthen the generalization claims.**
>
> **Response:** Thanks for the suggestion. To show the generalization of ID$^3$, we further run tests on RFW and obtain the following comparison results against the best SoTA model (IDiffFace) other than ID$^3$:
>
> | Models | African | Asian  | Indian | Caucasian | average |  std |
> |----------|----------|----------|----------|----------|----------| ----------|
> | IDiffFace | 71.10 | 75.83 | 78.68   | 82.05   | 76.91    | 4.64   |
> | ID$^3$ | 78.25| 79.23 | 84.05   | 85.28   | 81.70   | 3.48  |
>
> Note that RFW (Racial Faces in the Wild) is a comprehensive diverse benchmark for face recognition. It is notable for its diversity, capturing the facial variations within and across different racial groups, including but not limited to Caucasian, Black, Asian, and Indian individuals. It includes faces from different age groups, genders, and a variety of lighting and pose conditions, ensuring a realistic and challenging test bed for face recognition algorithms.
>
> From the table above, we observe that ID$^3$ outperforms IDiffFace by clear margins, which suggests its effectiveness and generalization to different groups of age, pose, ethnics, etc.
>
> **Weakness 2: It would be beneficial to have details on the computational demands and scalability of the model when deployed in practical, real-world scenarios.**
>
> **Response:** Thanks for the suggestion. The time complexity of our proposed Algorithm 3 is $O(mNT)$, where $N$ is the number of identities to be generated, $m$ is the number of images per identity and $T$ is the diffusion steps. The execution runtime of Algorithm 3 (for generating a face dataset that contains 500,000 face images with 10,000 identities) is 17.36 hours. This is a lot more efficient than manual effort which usually takes months to establish a dataset. The scalability of ID$^3$ is linear with $m$ and $N$, which suggests that ID$^3$ is promising in generating larger-scale datasets for synthetic face recognition when deployed in practical, real-world scenarios.
>
> **Question 1: What measures have been taken to ensure the model's robustness against diverse ethnic and age groups, given the model's reliance on identity embeddings?**
>
> **Response:** ID$^3$ models attributes in two ways. For those attributes that can be explicitly disentangled from identity, such as age and pose, ID$^3$ take explicit measures by treating age as one of conditioning signals, which renders ID$^3$ aware of age variations when generating ID-preserving faces accordingly. This makes ID$^3$ robust to diverse age groups. For those attributes that intertwine with identity, such as ethnics, ID$^3$ take implicit measures by automatically and flexibly capturing them through model training. Specifically, recall that ID$^3$ assumes access to the pretrained ArcFace, on the manifold of which dissimilar groups are separated and similar groups are gathered in terms of ethnics. Through end-to-end training, this characteristic acts as supervisory signals that are propagated back to the diffusion model of ID$^3$ (see Figure 1 in the main text). This makes ID$^3$ robust to diverse ethnic groups.
>
> **Question 2: Are there potential improvements or variations in the diffusion model that could further enhance identity preservation without sacrificing diversity?**
>
> **Response:** The proposed model ID$^3$ is inherently a conditional diffusion model conditioned on face attributes. Face attributes may include many aspects. Some can be explicitly disentangled from identity while others are implicitly intertwined with identity. Therefore, ID$^3$ models the former as explicit conditioning signals and models the latter as part of face image features. In this paper, for simplicity, we only show the effect of two major face attributes that are disentangled from identity: age and pose.  More such attributes can be added into our generation framework, such as expression. When more such attributes are incorporated into ID$^3$ explicitly, ID$^3$ becomes aware of these attributes upon which ID$^3$ utilizes the remaining information from face image features for better ID preservation enhancement.
>
> **Question 3: How does the model perform under constrained computational resources, and are there any strategies for optimizing its efficiency?**
>
> **Response:** The proposed model ID$^3$ is inherently a conditional diffusion model (DDPM) that normally costs much computational resources (e.g. ID$^3$ takes 21G GPU memory for inference; iDiffFace takes 24G GPU memory for inference). When deployed under constrained resources, one can use many efficient methods that aim to optimize DDPM inference efficiency such as DDIM [1], PLMSSampler [2] and SD (Spectral Diffusion) [3] while maintaining SFR performance.
>
>
> **References**
>
> [1] Song, Jiaming, Chenlin Meng, and Stefano Ermon. "Denoising diffusion implicit models." arXiv preprint arXiv:2010.02502 (2020).
>
> [2] Liu, Luping, et al. "Pseudo numerical methods for diffusion models on manifolds." arXiv preprint arXiv:2202.09778 (2022).
>
> [3] Yang, Xingyi, et al. "Diffusion probabilistic model made slim." Proceedings of the IEEE/CVF Conference on computer vision and pattern recognition. 2023.

---

### Author Rebuttal · Authors · 2024-08-06

We thank all reviewers for their time and effort in reviewing our work and their valuable comments about the paper. Attached is the one-page PDF that contains some figures and results for the rebuttal.

---

### Comment · Senior_Area_Chairs · 2024-08-14
**Further discussion with the authors**

Dear Reviewers,

Thank you for engaging in discussions with the authors. As the deadline for the discussion period is approaching, please provide any additional comments or feedback if you have any, to facilitate further discussion with the authors. Your input is valuable and appreciated.

Kind regards,

Your SAC

---

### Decision · Program_Chairs · 2024-09-25

**Decision:**

Accept (poster)

**Comment:**

To avoid privacy problem, this paper proposes ID3, an identity-preserving-yet-diversified diffusion model for generating synthetic face data for face recognition. The proposed method leverages identity embeddings and facial attributes to control inter-class and intra-class diversity of generated faces while preserving intra-class identity consistency, demonstrating promising performance on multiple synthetic face recognition benchmarks. Three out of four reviewers are/become positive to this work, while reviewer 37yP is not persuaded to become more positive. The main concern of 37yP lies in the evaluations, especially comparisons with other methods. It is not perfect, but enough to validate the effectiveness of the proposed method. So, a positive decision is reached. The authors should improve the manuscript by considering the review comments in the camera-ready version of the paper.

Ethics review comment: The authors should be mindful of the fact that using the FFHQ dataset for "development of improvement of facial recognition technologies" is discouraged and if possible, the authors should reconsider its usage in the camera-ready version.